# Admixture between old lineages facilitated contemporary ecological speciation in Lake Constance stickleback

David A. Marques [1,2,3,7], Kay Lucek [4,7], Vitor C. Sousa [3,5], Laurent Excoffier [3,6] & Ole Seehausen [1,2]

Ecological speciation can sometimes rapidly generate reproductively isolated populations coexisting in sympatry, but the origin of genetic variation permitting this is rarely known. We previously explored the genomics of very recent ecological speciation into lake and stream ecotypes in stickleback from Lake Constance. Here, we reconstruct the origin of alleles underlying ecological speciation by combining demographic modelling on genome-wide single nucleotide polymorphisms, phenotypic data and mitochondrial sequence data in the wider European biogeographical context. We find that parallel differentiation between lake and stream ecotypes across replicate lake-stream ecotones resulted from recent secondary contact and admixture between old East and West European lineages. Unexpectedly, West European alleles that introgressed across the hybrid zone at the western end of the lake, were recruited to genomic islands of differentiation between ecotypes at the eastern end of the lake. Our results highlight an overlooked outcome of secondary contact: ecological speciation facilitated by admixture variation.

[1] Aquatic Ecology and Evolution, Institute of Ecology and Evolution, University of Bern, Baltzerstrasse 6, CH-3012 Bern, Switzerland. [2] Department of Fish Ecology and Evolution, EAWAG Swiss Federal Institute of Aquatic Science and Technology, Center for Ecology, Evolution and Biogeochemistry, Seestrasse 79, CH-6047 Kastanienbaum, Switzerland. [3] Computational and Molecular Population Genetics, Institute of Ecology and Evolution, University of Bern, Baltzerstrasse 6, CH-3012 Bern, Switzerland. [4] Department of Environmental Sciences, University of Basel, Schönbeinstrasse 6, CH-4056 Basel, Switzerland. [5] Centre for Ecology, Evolution and Environmental Changes, University of Lisbon, Campo Grande 016, 1749-016 Lisbon, Portugal. [6] Swiss Institute of Bioinformatics, 1015 Lausanne, Switzerland. [7]These authors contributed equally: David A. Marques, Kay Lucek. Correspondence and requests for materials should be addressed to O.S. (email: ole.seehausen@eawag.ch)

Contemporary speciation studies have shown that speciation can sometimes be surprisingly fast even in the face of gene flow, allowing for sympatric divergence or persistence of incipient species. Many of these cases involve divergent natural selection or habitat-dependent sexual selection[1–4]. However, the origin of genetic variants underlying rapid speciation and reproductive isolation has remained unknown in all but a few cases[5–8], e.g., whether loci that contribute to reducing gene flow between populations are derived from de novo mutation, from standing genetic variation or from admixture variation acquired through introgression between divergent lineages. Understanding the process of speciation and its constraints requires knowledge on the origin of alleles under ecological, sexual or incompatibility selection. In turn, understanding the origin of alleles requires reconstructing the history of populations undergoing speciation.

We recently documented a case of contemporary ecological speciation in threespine stickleback (*Gasterosteus aculeatus* complex) of Lake Constance, Central Europe[3]. Lake- and stream-adapted ecotypes differ in predator defense and feeding morphology, ecology, nuptial coloration, migration behavior, and life history: lake stickleback grow larger than stream stickleback[9–12], are covered with larger or more lateral bony plates and possess longer spines as protection from predators[3,11–14], have longer gill rakers and jaws adapted to feeding on zooplankton instead of benthic invertebrates[9,12,13], migrate to lower reaches of streams to breed in contrast to resident stream stickleback[3] and start breeding 1 year later and die older than stream stickleback[9,12]. In a South-Eastern tributary of Lake Constance, both ecotypes breed in sympatry and they maintain phenotypic[10] and genomic differentiation despite ongoing gene flow[3]. Sympatric breeding of stickleback ecotypes is very rare and occurs here at a surprisingly early stage of speciation, given that historical records document the presence of stickleback in the Lake Constance catchment for the past 150 years only, and their prior absence from the basin[15–18]. The genomic architecture of this case is characterized by an undifferentiated genomic background interspersed by strong differentiation across multiple chromosomes, especially in low recombination regions and inversions enriched with quantitative trait loci (QTL) for divergent traits[3,19]. The lack of reduced diversity within genomic islands and of genomic background differentiation led us to hypothesize that ecotypes likely diverged in situ in Lake Constance, either from selection on standing genetic variation or admixture variation. From Lake Constance data alone, however, we were unable to distinguish these alternative origins of alleles.

A second study, including tributaries North and West of Lake Constance, has documented a similar genomic architecture but stronger genomic background differentiation in some parapatric lake vs. stream ecotype comparisons[14]. While standing genetic variation was also suggested as substrate for ecotype divergence, the study came to different conclusions regarding the mode and age of ecotype divergence. The authors estimated that ecotypes have been diverging for ~4500 generations, translating to ~9000 years or an early post-glacial divergence. Such a long time for ecological speciation corresponds to what has been reported in other well-studied cases of sympatric stickleback species[20], but is at odds with the historical ichthyological literature[15–18]. A mode of ecotype divergence termed 'ecological vicariance' was proposed in which a stream-adapted stickleback lineage colonized the streams of the area first, becoming isolated in different Lake Constance tributary streams due to the lake acting as a barrier for stream-adapted fish, followed by reconnection of populations once a lake-adapted ecotype had evolved from standing genetic variation[14].

Here, we re-evaluate the population history of Lake Constance stickleback and investigate the origin of genetic variants underlying contemporary ecological speciation and early persistence in sympatry. We place all previously studied lake and stream populations from the Lake Constance catchment in a wider European phylogeographic context, using new and published genome-wide single nucleotide polymorphism (SNP), mitochondrial, microsatellite and plate morph data[9,12,21,22]. We compare the fit of alternative demographic models to genomic data and estimate demographic parameters from the best-fitting model. The models we compare are: (1) primary divergence in situ, in which lake and stream ecotypes have recently diverged from standing genetic variation in a single lineage that colonized Lake Constance; (2) ecological vicariance (outlined above); (3) secondary contact, in which ecotypes correspond to West and East European lineages that have diverged in allopatry, recently met and sorted between lake and stream habitats with exchange of genes at the lake-stream boundaries; and (4) hybrid origin, in which one of the ecotypes has recently evolved through hybridization between divergent West and East European lineages or sorting of admixture variation following introgression from one divergent lineage into the other lineage.

Our analyses reveal admixture variation as genetic source of ecotype differentiation and thus contemporary ecological speciation, derived from hybridization between two divergent lineages from at least two European watersheds. We find that stream ecotypes in Lake Constance show a gradient of admixture between divergent European stickleback lineages: near the zone of secondary contact, stream ecotypes show a 50:50 hybrid origin, while far from the zone of contact, novel stream ecotypes evolved in situ from predominantly one genomic background, aided by sorting of introgressed alleles. Our analyses provide evidence for the hypothesis that admixture variation can be an important facilitator of rapid ecological speciation and adaptive radiation[23–25] and uncover an unexpected outcome of secondary contact: repeated ecological speciation beyond the contact zone. Our results imply that caution should be taken when inferring modes and times of speciation from population genome sequence data in the absence of a thoroughly sampled phylogeographic context.

## Results

**The origin of Lake Constance stickleback.** We identified five different mitochondrial DNA (mtDNA) haplotypes in the Lake Constance catchment (Fig. 1b), all of them matching haplotypes that today can be found in four main European river catchments, draining respectively into the Baltic Sea, the North Sea, the Mediterranean Sea, and the Black Sea[12,22,26] (Fig. 1a). All five mtDNA haplotypes are part of central European lineages, highly divergent from Mediterranean and Black Sea lineages[26–29] (Fig. 1a, b). Phylogenomic analyses based on concatenated genome-wide SNPs derived from *SbfI*-restriction-site-associated DNA (RAD) sequencing confirm the position of Lake Constance stickleback among the central European lineages (Fig. 1c, Supplementary Fig. 1). Among these are at least two old sublineages from East and West Europe that represent different nominal species: the West European *Gasterosteus gymnurus* lacking lateral bony plates except for the structural plates and the fully plated East and North European *Gasterosteus aculeatus* that also resembles the marine form distributed along the North European coasts[30,31] (Fig. 1d). In Lake Constance, fully, low-, and partially plated stickleback occur (Fig. 1e). Thus, mtDNA haplotypes suggest, consistent with phenotypes, contributions from multiple old central European stickleback lineages to the Lake Constance catchment[26–28].

Importantly, contributions are not equally distributed among populations and ecotypes in the Lake Constance catchment (Fig. 1b). Lake and stream populations from South-Eastern

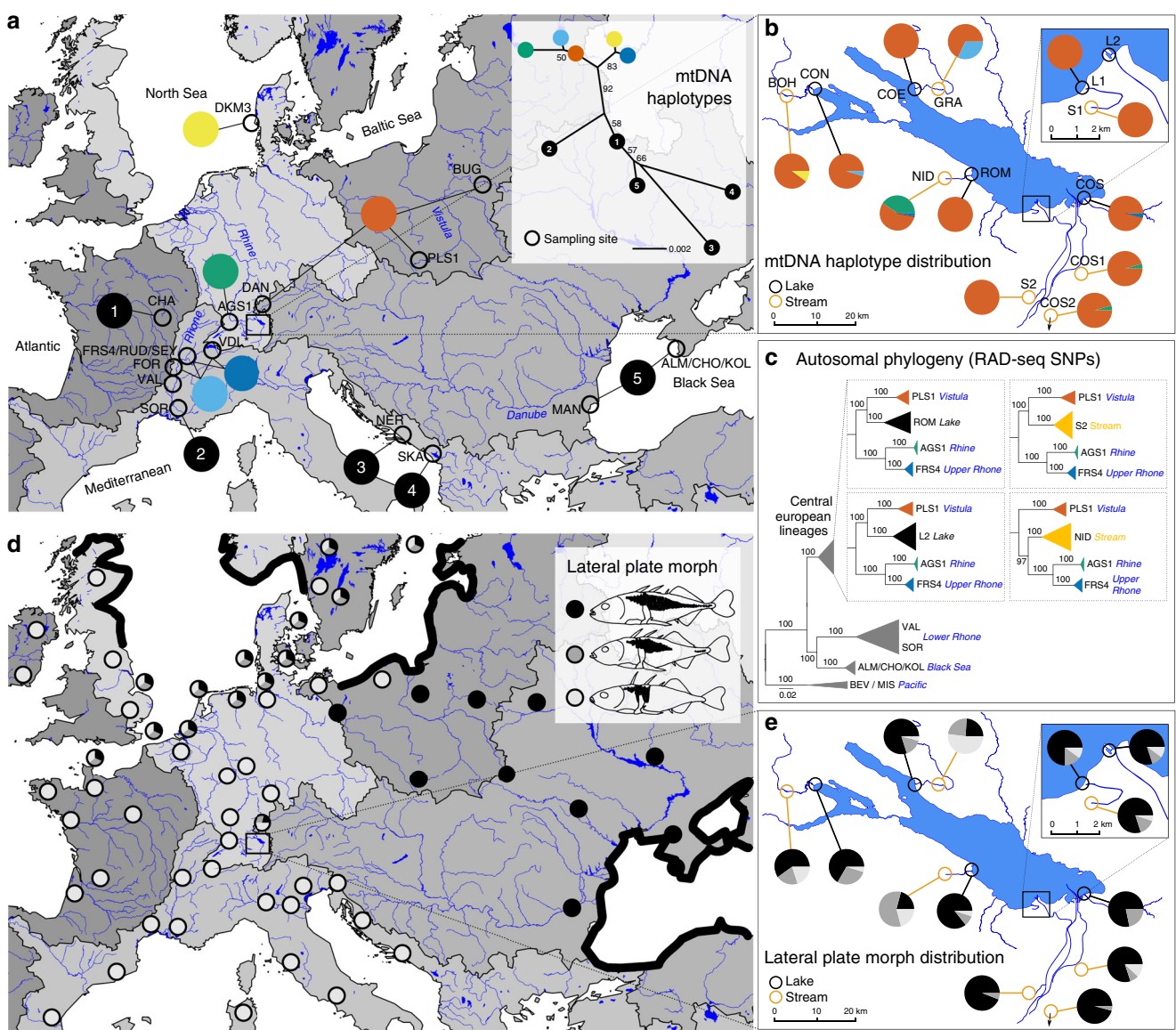

**Fig. 1** Phylogeography of Lake Constance stickleback. Stickleback from the lake and South-Eastern streams show a predominantly East European origin in mtDNA haplotype, phenotype and genome-wide SNPs. In contrast, stickleback from streams West and North of the lake show signs of admixture with West European lineages from the Rhine and upper Rhone. **a** European distribution and maximum-likelihood phylogeny of mtDNA haplotypes found in Lake Constance (colored circles) and in neighboring watersheds (black circles). All Lake Constance haplotypes are part of the 'Northern European clade' *sensu*[26,27], highly divergent from all haplotypes known from the Mediterranean and Black Sea basins (numbered 2–5). Branch labels show bootstrap support in percent, see Methods for haplotype names. **b** Distribution of mtDNA haplotypes in Lake Constance. **c** Maximum-likelihood phylogenies of concatenated autosomal SNPs embedding single Lake Constance populations into the same European phylogeny consisting of divergent clades as shown in (**a**). Branch labels show bootstrap support in percent. See Supplementary Fig. 1 for a tree incorporating all Lake Constance populations together. **d** Distribution of lateral plate phenotypes across Europe pre-1963, redrawn from Munzing[30] with information for additional contemporary populations added. **e** Distribution of lateral plate phenotypes across the Lake Constance basin. Note the high prevalence of low- or partially plated individuals in streams West and North, but not South-East of the lake. Watershed maps are derived from "Water Base: Global River Basins" by The World Bank used under CC BY 4.0, river and lake maps from "European catchments and Rivers network system (Ecrins)" by the European Environment Agency (EEA). Source data are provided as a Source Data file

tributaries to Lake Constance are nearly fixed for an mtDNA haplotype found in East Europe in the Vistula and Upper Danube catchments (Fig. 1b). The nuclear genome of these populations also resembles East European stickleback and phylogenetic analyses with genome-wide SNPs cluster them as sister lineages (Fig. 1c). Most individuals are fully plated, while partially or low-plated individuals occur at low frequency (Fig. 1e), consistent with an East European freshwater or a marine origin of stickleback from Lake Constance and its South-Eastern tributaries.

In contrast, some stream populations from Northern and Western tributaries of the lake appear to be of hybrid origin between West and East European stickleback lineages, meeting in a secondary contact zone where the former tend to inhabit streams and the latter the lake. West European haplotypes that are known otherwise only from the Rhine, the upper Rhone and the southern North Sea occur at high frequency in populations GRA and NID (Fig. 1a, b, see Supplementary Table 1 for population abbreviations). A phylogenetic analysis based on concatenated genome-wide autosomal SNPs derived from

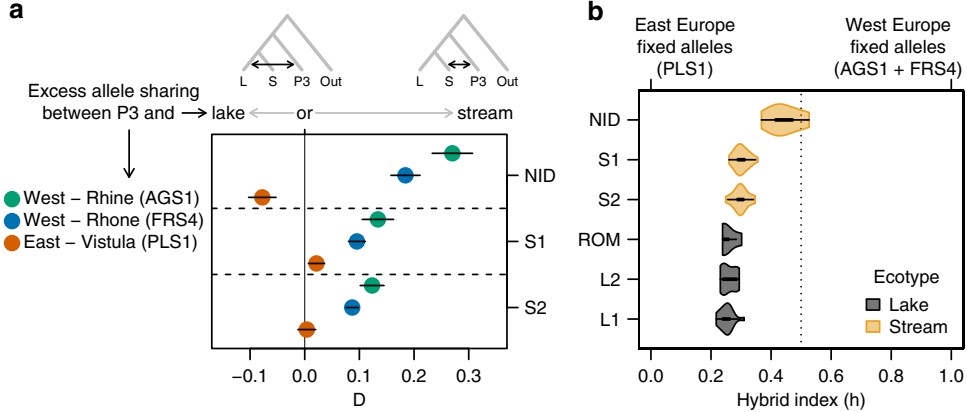

**Fig. 2** Evidence for admixture between West and East European lineages in Lake Constance. **a** The D-statistic shows significant excess allele sharing between two West European lineages (P3: upper Rhone = FRS4, Rhine = AGS1) and stream stickleback in three streams (S: S2, S1, NID) relative to lake stickleback (L: L1). The lake population L1 shows excess allele sharing with the East European sister lineage (P3: Vistula = PLS1) only when compared with the stream population NID, suggesting that L1, S2, and S1 contain similarly large proportions of East European ancestry. Error bars indicate ± 3 standard deviations around D-estimates. Japan Sea stickleback *Gasterosteus nipponicus* was used as outgroup ('Out'), see Supplementary Fig. 3 for near-identical results with other Constance lake populations or outgroups. **b** Admixture proportions of six different Lake Constance stickleback populations estimated from SNPs that are fixed between East and West European stickleback populations (PLS1 vs. FRS4 + AGS1, n = 299 SNPs each spaced at least 100 kb apart). A hybrid index of 0 implies that all 299 loci are fixed for the East European allele, 1 implies fixation for the West European alleles. Black box plots delineate the 1st and 3rd quartile, with error bars extend these by max. 1.5 times the interquartile range. Source data are provided as a Source Data file

*SbfI*-RAD sequencing clusters the stream population NID with West European lineages rather than with East European lineages, with very high bootstrap support (Fig. 1c). However, when all Lake Constance lake and stream populations are included into the phylogeny, the stream population NID clusters with the other Lake Constance samples as sister of the East European lineage, with slightly reduced bootstrap support (Supplementary Fig. 1), as expected for a population of hybrid origin[23]. Phenotypically, the two stream populations GRA and NID are dominated by low and partially plated stickleback, respectively, in contrast to the fully plated stream populations from South-Eastern tributaries and the lake population (Fig. 1d).

We assessed the presence and extent of admixture between West and East European lineages in the Lake Constance catchment with D-statistics, hybrid index and clustering analyses, using a *SbfI*-RAD sequencing SNP dataset (see Methods). When Constance and East European populations are treated as sister lineages in a phylogenetic tree, all Lake Constance populations show a significant excess of derived allele sharing with West European populations from the Rhine and the upper Rhone (Supplementary Fig. 2), confirming admixture between West and East European lineages in Lake Constance. The stream population NID shows the strongest signal for the D-statistic, the other stream populations (S1, S2) also show D-statistics significantly different from zero and two lake populations have marginally significant D-statistics (L1, L2, but not ROM, Supplementary Fig. 2), indicating varying extents of admixture with a West European lineage. Overall, populations of the stream ecotype contain more West European alleles than the lake ecotype populations: with lake and stream populations as sisters, all stream populations showed excess allele sharing with West European lineages, with a particularly strong signal in NID (Fig. 2a, Supplementary Fig. 3). Only when contrasted with NID as the stream population, lake populations show excess allele sharing with East European lineages (Fig. 2a, Supplementary Fig. 3), in agreement with a hybrid origin of the NID population.

We estimated admixture proportions by computing the hybrid index from 299 divergently fixed SNPs between the East and West European populations (PLS1 vs. (FRS4, AGS1), Fig. 2b). Admixture estimates for individuals from population NID ranged

from 36–53% West European origin with a mean of 44%, supporting a hybrid origin of this population. Lake and stream populations from South-Eastern tributaries showed lower individual admixture proportions of 21–36% (Fig. 2b), with the stream ecotype (29–30%) showing a slightly elevated hybrid index compared with the lake ecotype (25–26%, Fig. 2b). In a clustering analysis of SNP and microsatellite data (Supplementary Figs. 4, 5), the Lake Constance stream population GRA even showed a predominantly West European origin while the remaining Lake Constance lake and stream populations showed minor admixture from West European origin.

**Contrasting modes of ecotype divergence in Lake Constance.** We compared different modes of stickleback ecotype divergence in Lake Constance using a coalescent demographic modeling framework, and taking advantage of the West and East European sister lineages as discussed above. We tested the fit of multiple neutral demographic models to the observed site-frequency spectra (SFS) computed from sites in high recombination rate regions in order to minimize effects of selection on the SFS[32]. We compared models on three hierarchical levels of complexity (Fig. 3). First, we established the relationships among the three allopatric European stickleback lineages contributing to Lake Constance stickleback using three population models and three-dimensional (3D) SFS (Fig. 3a). This allowed us to estimate split times unaffected by gene flow in Lake Constance at low model and data complexity. Then, we increased complexity by adding single Constance lake or stream populations to the model in order to quantify major (Fig. 2b) and minor (Fig. 2c) contributions of the European lineages in four population models optimized on joint two-dimensional (2D) SFS for all population pairs. Finally, we compared different modes of incipient speciation: primary divergence of ecotypes, ecological vicariance, persistence in secondary contact and hybrid origin, by including one lake/stream population pair each into five population models optimized on joint 2D-SFS (Fig. 2d). We optimized model parameters on observed SFS from two RAD sequencing sets, an *SbfI*-RAD dataset[3,33] including European sister lineages and lake (L2) and stream (S2, NID) populations, and an *NsiI*-RAD dataset[14]

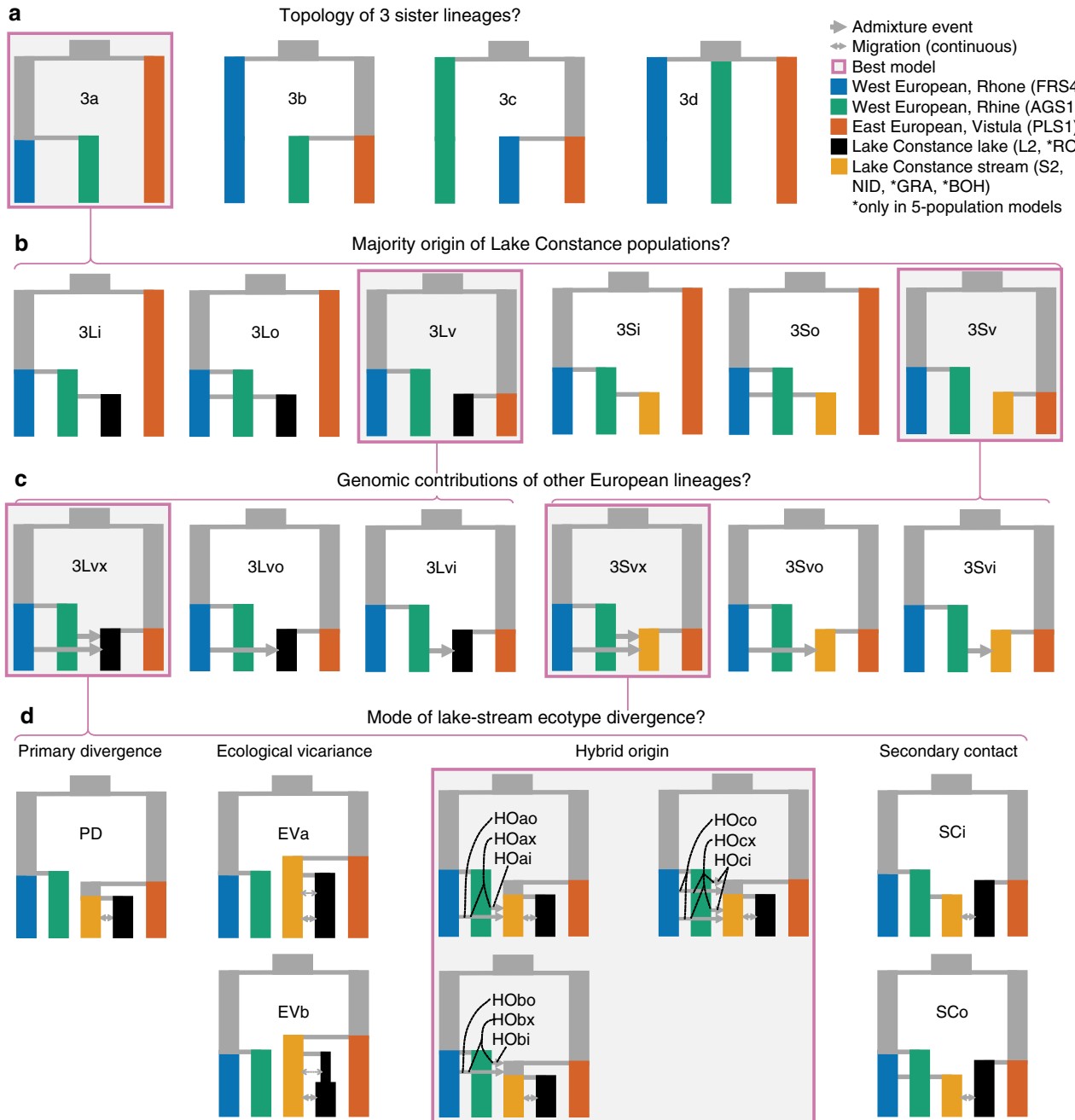

**Fig. 3** Hierarchical demographic modeling approach for ecotype divergence. **a** We first established relationships among three allopatric European lineages from the Rhine, the upper Rhone and the Vistula rivers. The model 3a grouping populations into a West (Rhine, upper Rhone) and an East European (Vistula) clade was best supported by the data. **b** Next, we tested from which of the three lineages the majority of the genome of Lake Constance populations (L2, S2, NID) was derived. The best supported models suggest a majority contribution from the East European lineage for all populations (3Lv, 3 Sv). **c** Then, we tested whether other lineages contributed to Lake Constance populations (L2, S2, NID), with a model of contributions from both West European lineages best supported for all populations (3Lvx, 3Svx). **d** Finally, we compared different modes of ecotype divergence: primary divergence in situ, ecological vicariance, secondary contact and hybrid origin. For all pairwise lake-stream comparisons, a hybrid origin model clearly outperformed the other modes of divergence (Supplementary Fig. 8, Table 1). Pink rectangles highlight the best supported models. Little letters in model names denote models with admixture from West European lineages Rhone ('o'), Rhine ('i') or from both ('x'). Source data are provided as a Source Data file

including Constance lake (ROM) and stream (NID, GRA, BOH) populations (see Methods).

We infer West and East European lineages to have split ~3665 (95% confidence interval (CI): 3636–4877) generations ago, while the upper Rhone and Rhine populations split 1710 (1710–2239) generations ago according to our best 3-population model (Fig. 3a, Supplementary Fig. 6, Table 1). Admixture/gene flow between allopatric European lineages did not significantly

improve the model fit, supporting that the three populations we use (FRS1, AGS1, PLS1) are non-admixed (Supplementary Figs. 7, 8, Supplementary Table 2). Adding any single Lake Constance population (L2, S2, NID) to the trio of allopatric European lineages revealed that the respective best supported models all suggest that the majority of their genomes is of East European origin (Fig. 3b, Table 1). Likewise, models with additional contributions from both West European populations (Rhine,

**Table 1 Fit of demographic models to the observed data**

| Dataset | Sbfl(1–3) | | Dataset | Sbfl(1–4) | | Dataset | Sbfl(1–3,6) | | Sbfl(1–3,7) | |
|---|---|---|---|---|---|---|---|---|---|---|
| Model | ΔLL | ΔAIC | Model | ΔLL | ΔAIC | Model | ΔLL | ΔAIC | ΔLL | ΔAIC |
| 3a* | 79 | −7* | 3Lo | 1343 | −5313 | 3So | 1145 | −4435 | 978 | −1963 |
| 3b | 581 | −2317 | 3Li | 1513 | −6095 | 3Si | 1257 | −4953 | 1012 | −2121 |
| 3c | 598 | −2398 | 3Lv | 600 | −1892 | 3Sv | 578 | −1826 | 968 | −1918 |
| 3d | 503 | −1959 | 3Lvx* | 191 | −14* | 3Svx* | 181 | −4* | 551 | −2* |
| | | | 3Lvo | 222 | −153 | 3Svo | 197 | −73 | 573 | −105 |
| | | | 3Lvi* | 188 | 0* | 3Svi* | 181 | 0* | 550 | 0* |

| Dataset | Sbfl(1–4,6) | | Sbfl(1–4,7) | | Nsil(1–3,5,7) | | Nsil(1–3,5,9) | | Nsil(1–3,5,8) | |
|---|---|---|---|---|---|---|---|---|---|---|
| Model | ΔLL | ΔAIC | ΔLL | ΔAIC | ΔLL | ΔAIC | ΔLL | ΔAIC | ΔLL | ΔAIC |
| PD | 650 | −1063 | 1532 | −1304 | 184 | −313 | 154 | −249 | 411 | −789 |
| EVa | 642 | −1028 | 1535 | −1321 | 185 | −318 | 154 | −253 | 414 | −809 |
| EVb | 647 | −1053 | 1528 | −1289 | 182 | −309 | 152 | −244 | 424 | −853 |
| HOax* | 437 | −89 | 1421 | −799 | 159 | −205 | 104 | −25* | 306 | −313 |
| HOai* | 440 | −101 | 1420 | −790 | 160 | −205 | 99 | 0* | 343 | −483 |
| HOao | 461 | −199 | 1428 | −827 | 175 | −278 | 124 | −117 | 348 | −503 |
| HObx* | 424 | −29* | 1337 | −411 | 116 | −7* | 142 | −199 | 435 | −907 |
| HObi* | 418 | 0* | 1337 | −410 | 117 | −9* | 138 | −178 | 440 | −929 |
| HObo* | 453 | −160 | 1355 | −491 | 122 | −31* | 144 | −208 | 435 | −907 |
| HOcx* | 421 | −20* | 1249 | −12* | 113 | 0* | 101 | −18* | 237 | 0* |
| HOci* | 417 | −1* | 1247 | 0* | 119 | −20* | 129 | −141 | 279 | −189 |
| HOco* | 454 | −170 | 1283 | −166 | 122 | −37* | 142 | −203 | 246 | −41* |
| SCi | 711 | −1346 | 1436 | −860 | 509 | −1811 | 236 | −626 | 776 | −2472 |
| SCo | 784 | −1682 | 1529 | −1288 | 434 | −1465 | 237 | −631 | 551 | −1435 |

Best-fitting models and models with very similar likelihood are marked with an asterisk (*). Shown are log$_{10}$ likelihood differences between observed and expected site-frequency spectra (ΔLL) and difference in Akaike information criterion (ΔAIC) between the best and all models for a given dataset. For Nsil-data, the three West European lineages were modeled as unsampled ('ghost') populations, with parameters for the latter fixed to best estimates of model 3a (Supplementary Fig. 6). Numbers in brackets indicate populations used: West (Rhone: 1:FRS4, Rhine: 2:AGS1) and East European (3: PLS1), Lake Constance lake (4:L1, 5:rROM) and stream populations (6:S2, 7:NID, 8:GRA, 9:BOH). Source data are provided as a Source Data file.

upper Rhone) were best supported for the lake and both stream populations (Fig. 3c, Table 1). Strikingly, estimates suggest that the lake population received lower contributions from West Europe (Rhine: 6.7%, upper Rhone: 0.7%) than the two stream populations S2 (Rhine: 16.6%, upper Rhone: 0.1%) and NID (Rhine: 27.2%, upper Rhone: 0.2%), in line with D-statistic and hybrid index analyses.

The mode of ecotype divergence 'hybrid origin' fits the observed data considerably better than models of primary divergence in situ, ecological vicariance or of secondary contact (Fig. 3d, Table 1, Supplementary Fig. 8). This is true for all lake-stream comparisons tested with two independent and partially overlapping RAD sequencing datasets: S2 vs. L2 and NID vs. L2 using SbfI-derived RAD sequencing data and NID vs. ROM, GRA vs. ROM, and BOH vs. ROM using NsiI-derived RAD sequencing data. Hybrid origin models best capture the predominantly East European genomic background of all Lake Constance populations and higher allele sharing of stream ecotype populations with West European stickleback lineages (Fig. 4b, c). In the best supported models, West European lineages contributed most to stream ecotype populations in Northern and Western tributaries of Lake Constance (NID: 31%, BOH: 32%, GRA: 48% West European contribution) with lower contributions to the respective lake ecotype (Fig. 4b, c). In contrast, in the South-Eastern tributaries of Lake Constance the ancestor of both lake and stream ecotypes received 14% (6.9–14%) of West European alleles, consistent again with the hybrid index results.

In our models, lake and stream ecotype divergence time estimates vary considerably for different lake-stream comparisons: 408 (95% CI: 382–680) generations for S2, 1276 (1015–1446) generations for NID, 2819 (1626–2819) generations for BOH and 1065 (888–1232) generations for GRA (Fig. 4b, c). Effective population size estimates for the lake population (L2, ROM) varied from $2N_e = 3713$ to 8473 (Fig. 3b, c), while effective

stream population sizes were estimated to be smaller in one case (S2), similar to lake population sizes in two cases (NID, BOH) and larger in one case (GRA, Fig. 3b, c).

In summary, demographic modeling suggests that West and East European lineages diverged ~4000 generations ago, translating to ~8000 years assuming 2 years average age of reproduction as generation time. After several thousand generations of isolation, these West and East European lineages met in the Lake Constance system, perhaps at lake-stream boundaries North and West of the lake. Stream stickleback there are of hybrid origin, having received most of their genome from a West or East European lineage depending on the population. Introgression across the secondary contact zone contributed some West European alleles to the lake ecotype and also to stream ecotypes in South-Eastern tributaries.

**Admixture variation fueled incipient speciation.** Our analyses reveal that stickleback in Lake Constance and South-Eastern tributaries originate from a phenotypically 'marine'-like, fully plated East European freshwater lineage (Gasterosteus aculeatus), while tributaries North and West of the lake are of hybrid origin with the 'classical freshwater'-type, low-plated stickleback from West European streams (Gasterosteus gymnurus). Even though stream ecotypes differ in admixture proportions and plate morph distributions (Fig. 1e), all stream ecotypes from around Lake Constance differ consistently and in parallel direction from the lake ecotype in many other traits such as body size[9,12], body shape[11,21], gill raker length[11,21], age of first reproduction[9,12], pelagic vs. benthic foraging efficiency[9,12,34], with some of them being heritable[10]. This raises the question: was ecotype divergence in South-Eastern tributaries of Lake Constance facilitated by admixture variation arising from hybridization between old West and East European stickleback lineages?

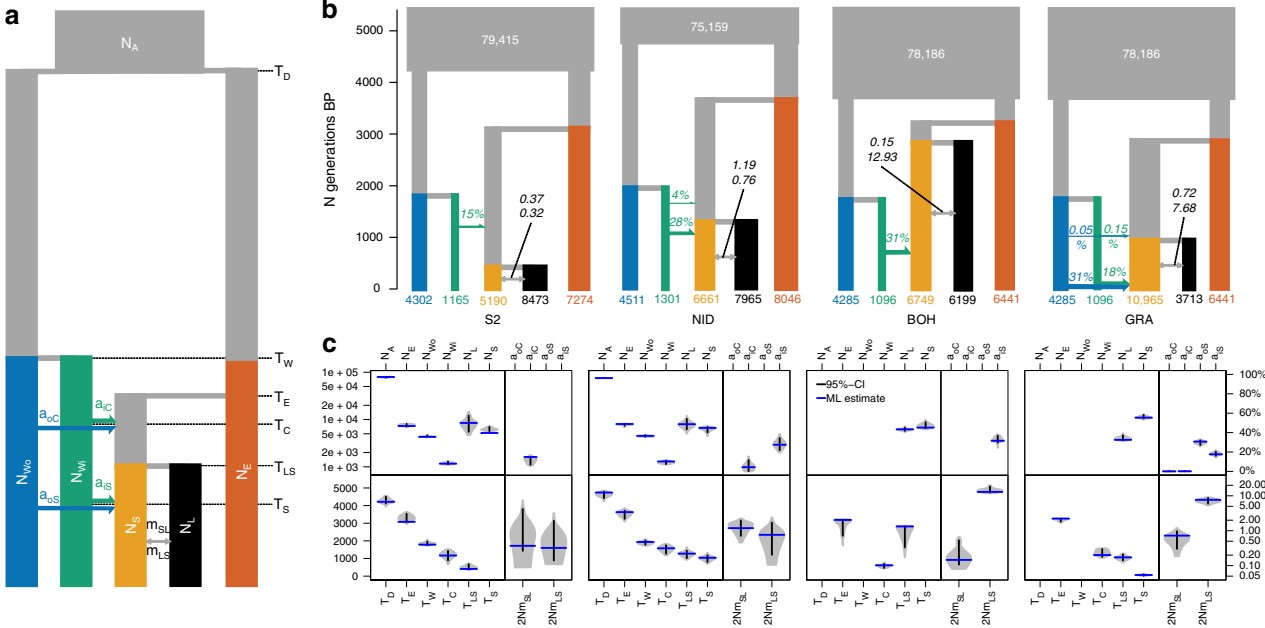

**Fig. 4** Demographic history of Lake Constance stickleback. **a** Model parameters in hybrid origin models for population size ($N_i$) in units of $2N_e$, admixture proportion ($a_{ij}$) in percentage of the target population, migration rate ($m_{ij}$) in units of $2N_e$ migrants per generation (forward in time), and time ($T_i$) in numbers of generations. **b** Maximum-likelihood parameters for the best models selected for the lake-stream pairs L2-S2, L2-NID, ROM-BOH, and ROM-GRA. All models are represented in the same time scale and the width of lineages is proportional to the effective size of populations (indicated inside or below each population). Unidirectional arrows indicate admixture events, bi-directional arrows indicate migration, with values above giving the number of migrants forward in time in $2N_e m$ units. **c** Block-bootstrap parameter confidence intervals for the best models outlined in (**b**), with maximum-likelihood (ML) parameter estimates indicated in blue and black error bars showing 95%-confidence intervals (95%-CI). Source data are provided as a Source Data file

We seek the answer to this question by identifying the origin of alleles in genomic islands of differentiation between lake and stream ecotypes breeding in sympatry. In a previous study[3], we identified 19 genomic islands with unexpectedly high differentiation between lake and stream ecotypes and parallel allele frequency changes in two South-Eastern tributaries to Lake Constance, one tributary with sympatric (S1) and one with parapatric reproduction (S2). These genomic islands are thus candidate loci for reproductive isolation or ecological adaptation to either the lake or the stream habitat due to their persistence in sympatry, parallel allele frequency shifts, habitat associations and their enrichment with QTL controlling adaptive traits divergent between ecotypes[3]. Before identifying the origin of alleles in genomic islands, we aim to confirm that these genomic islands represent regions that resist gene flow between ecotypes due to either divergent selection or reproductive isolation, rather than representing regions that diverged due to background selection[35].

A rich literature has shown that heterogeneous differentiation ($F_{ST}$) across the genome will arise as by-product of background selection in the absence of gene flow[35–41]. Background selection removes proportionally more linked variation in low recombination regions, leading to negative correlations of differentiation ($F_{ST}$) with recombination rate, absolute divergence ($d_{XY}$) and diversity levels ($\pi$), and correlated genome-wide differentiation across populations and taxa[35–41]. Allopatric stickleback populations from different European watersheds lack gene flow, but we find no correlation of $F_{ST}$ with recombination rate or absolute divergence, even though differentiation landscapes are weakly correlated and $F_{ST}$ and diversity show negative correlations in some populations (Supplementary Fig. 9). The lack of association with recombination rate suggest that background selection might not be a major driver of genome-wide differentiation ($F_{ST}$)

between allopatric European stickleback populations, even in the absence of gene flow.

In contrast to those, Lake Constance ecotypes breed either in sympatry or parapatry with ample opportunity for gene flow, or in allopatry between different streams with limited gene flow between them. We find negative correlations between $F_{ST}$ and recombination rate among sym- and parapatric lake and stream ecotypes, but not between allopatric stream populations around Lake Constance (Supplementary Fig. 9). However, neither differentiation landscapes between lake-stream ecotypes or between allopatric streams are correlated with diversity or absolute divergence, nor are they with allopatric differentiation landscapes from outside Lake Constance, both genome-wide (Supplementary Fig. 9) and in genomic islands[3]. These combined patterns are best explained by a scenario of divergent selection where lake and stream ecotype differences persist against gene flow aided by low recombination, rather than by background selection.

If admixture variation facilitated ecotype divergence in South-Eastern tributaries of Lake Constance, we expect differential sorting of West- and East-derived alleles in genomic islands of differentiation between lake and stream ecotypes. We thus compared genome-wide differentiation and differentiation in genomic islands between Constance lake/stream stickleback and West European populations (Fig. 5b, c, Supplementary Figs. 10, 11). Genome-wide, both ecotypes from South-Eastern Lake Constance differ strongly from West European populations (weighted mean $F_{ST (L1 \text{ vs. } AGS1)} = 0.50$, $F_{ST (S1 \text{ vs. } AGS1)} = 0.46$, $F_{ST (S2 \text{ vs. } AGS1)} = 0.48$, Fig. 5b) and are more similar to East European populations ($F_{ST (L1 \text{ vs. } PLS1)} = 0.31$, $F_{ST (L1 \text{ vs. } PLS1)} = 0.30$, $F_{ST (S2 \text{ vs. } PLS1)} = 0.33$), consistent with a largely East European genomic background of both ecotypes. In genomic islands of ecotype differentiation, however, stream stickleback are

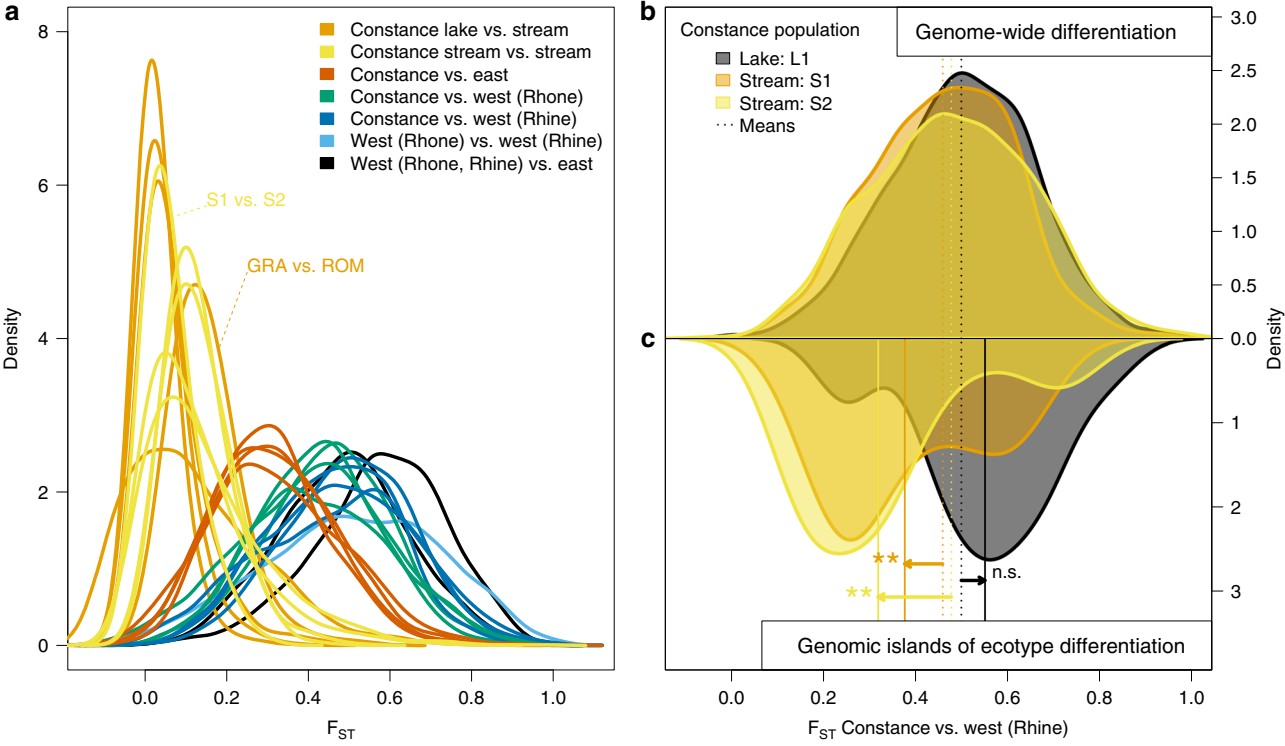

**Fig. 5** Admixture variation fueled ecological speciation. Genomic islands of parallel ecotype differentiation[3] between Lake Constance lake and stream ecotypes from South-Eastern tributaries[3] originate from admixture variation contributed by divergent West and East European lineages. **a** Genome-wide allopatric differentiation ($F_{ST}$) levels between West and East European lineages (black lines) and between Lake Constance and West European populations are similar (green/dark blue), while differentiation is much lower between Lake Constance and East European comparisons (red). Shown are pairwise $F_{ST}$ distributions between lake (L1, ROM), stream (S1, S2, BOH, GRA, NID), West (AGS1, FRS4), and East (PLS1) European populations. **b**, **c** Despite strong genome-wide differentiation (**b**) of both ecotypes from West European populations, the stream ecotype strongly resembles the West European lineage in genomic islands of parallel ecotype differentiation[3], in contrast to the lake ecotype (**c**). Shown are $F_{ST}$ distributions for lake or stream (L1, S1, S2) versus West European lineage (AGS1). All $F_{ST}$ distributions are distributions of weighted mean $F_{ST}$ computed in windows of at least 2500 sequenced base pairs. Arrows/stars indicate genomic island means (full lines) significantly different from genome-wide means (dashed lines, permutation test alpha levels 5% = * and 1% = **). Source data are provided as a Source Data file

significantly less differentiated from West European stickleback ($F_{ST\ (\text{islands, S1 vs. AGS1})} = 0.38$, two-sided permutation test $p < 0.01$, $F_{ST\ (\text{islands, S2 vs. AGS1})} = 0.32$, $p < 0.01$) than genome-wide, in contrast to the lake ecotype for which both genomic islands and genomic background show similar levels of differentiation from West European stickleback ($F_{ST\ (\text{islands, L1 vs. AGS1})} = 0.55$, Fig. 5c, Supplementary Table 3). Genomic islands between ecotypes thus originated from re-assortment of West- and East-derived alleles into lake and stream ecotypes. West-derived alleles are predominantly found in the stream ecotype (Fig. 6a–c). Only a smaller number of genomic islands show sorting into the opposite direction, with the East-derived allele at higher frequency in the stream ecotype and the West-associated allele at high frequency in the lake ecotype (e.g., genomic islands on chromosomes VII and XII, Fig. 6d). Stream ecotype stickleback in South-Eastern tributaries of Lake Constance thus do not represent a mere reassembly of a West European lineage stickleback, but rather a novel genomic combination.

## Discussion

Here we have demonstrated that secondary contact between divergent lineages and the re-assortment of introgressed alleles into ecotypes underlie recent ecological speciation across lake-stream habitat boundaries in Lake Constance. Our analysis reveals an unexpected outcome of secondary contact between old allopatric lineages: rapid in situ ecological speciation outside the secondary contact zone fueled by introgression of admixture variation beyond the hybrid zone. It also explains and reconciles contrasting conclusions of previous studies regarding the origin, age and mode of ecotype divergence in Lake Constance. Finally, our results raise interesting questions about the evolutionary potential arising from recombining old alleles into new combinations and how distinctive the new combinations might be from parental combinations.

Analyzing the incipient radiation of Lake Constance stickleback in a European-wide biogeographic and phylogeographic context revealed contributions of at least two old central European lineages of freshwater stickleback that had evolved in isolation for several thousand generations before coming into secondary contact. Secondary contact has not been invoked previously in this system and explains the old, early Holocene (post-glacial) divergence times that had been estimated between Lake Constance ecotypes by applying a primary divergence model representing an ecological vicariance scenario[14]. Our analyses of mitochondrial, microsatellite, and genome-wide SNP data in a larger phylogeographic context using an array of different methods such as phylogenetic reconstruction, demographic modeling, D-statistics, hybrid index, and cluster analysis clearly reject an origin from a single lineage and identify more than one source of origin. The colonization of the Lake Constance catchment by multiple lineages is consistent with historical accounts reporting introductions of stickleback into streams of North-Western Lake Constance between 1920 and 1940, in which the

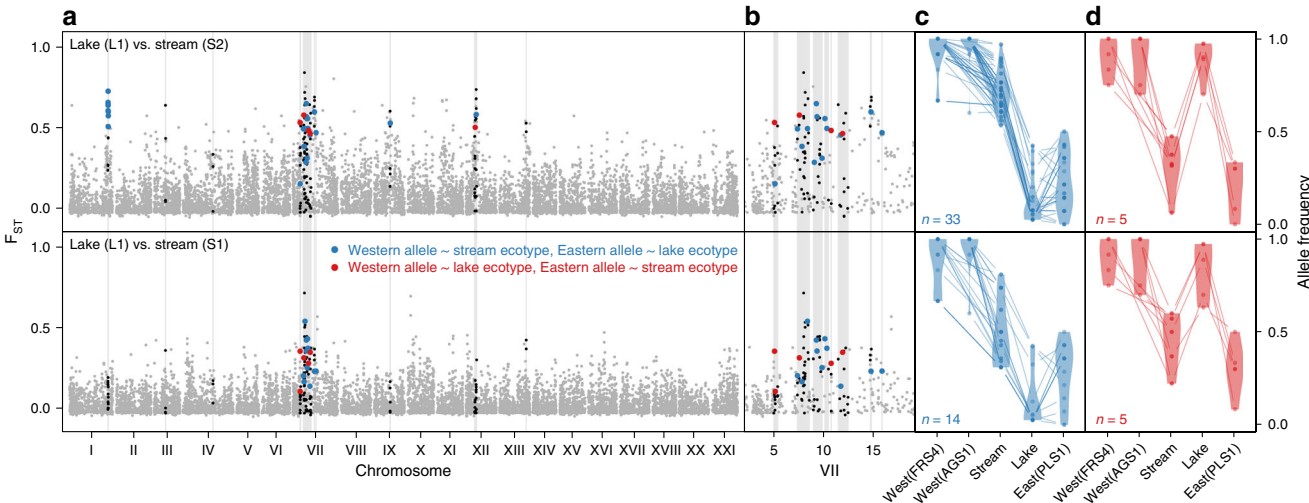

**Fig. 6** New combinations of old West and East European alleles made lake and stream ecotypes. Stream stickleback from South-Eastern tributaries of Lake Constance represent new combinations of West- and East-derived alleles and not a simple reassembly of a West European stickleback genotype.
**a**, **b** Genomic differentiation between lake and stream stickleback from South-Eastern tributaries of Lake Constance, including stream 1 where ecotypes breed in sympatry (S1). Genomic islands of elevated differentiation are highlighted in light gray backgrounds, with SNPs shown in black, blue or red. Blue SNPs indicate lake-stream differentiated SNPs where the lake ecotype carries the East European allele and the stream ecotype the West European allele. Red SNPs indicate the opposite pattern—where the stream ecotype carries an East European allele and the lake ecotype a West European allele. Ancestry assignment of alleles is based on strong allele frequency differences between West (AGS1 + FRS4) and East (PLS1) European populations (panels **c**, **d**). **a** Genome-wide differentiation. **b** Magnification of chromosome VII. **c**, **d** Most lake-stream differentiated SNPs show the blue pattern, but red pattern SNPs make a significant proportion. Source data are provided as a Source Data file

introduction of both fully plated and low-plated individuals was documented[17]. This is additional to historical accounts of introductions from unknown sources into streams South of Lake Constance in the late 19th century[16]. Consistent with an earlier mitochondrial phylogeography[22] and the biogeographic context of a highly plated freshwater phenotype[30,42], our analyses based on genome-wide SNPs identified East Europe as the major source of origin. The same lineage is also found in the nearby upper Danube drainage[12] and it is one of the central European lineages[26–28] as opposed to the highly divergent Black Sea or Mediterranean lineages. Such a central European rather than Black Sea origin is also in agreement with the historical ichthyologic record, which documents the absence of stickleback from the upper Danube until the early 20th century[15,18] and thus rather a more recent colonization of the upper Danube and the Lake Constance basin from Northern catchments aided by human introductions instead of a natural colonization from the Black Sea.

Our demographic models were not able to fully resolve the discrepancy between historical absence of stickleback in Lake Constance until 1860–70[16,17] and ecotype divergence time estimates hundreds of generations ago. Our parameter estimates are still associated with high uncertainty, as reflected by their wide confidence intervals in each model and variation for the same parameters between different datasets (Fig. 4). The estimated divergence times, times of introgression (which are notably more recent for two streams) and amounts of introgression however may lead to multiple plausible estimates in parameter space. Selection, in particular on old variation as we document here, may have further biased these estimates toward older times, despite our attempts to exclude sites under selection by considering only high recombination region SNPs for the SFS computation. The rather sparse *SbfI*-RAD sequencing data on one hand and the lack of sister lineage data for denser *NsiI*-RAD sequencing data on the other hand may have limited our power to better estimate these parameters. Further model optimization

incorporating additional variation in population sizes, e.g., bottlenecks and expansions, in combination with higher resolution SNP data may be able to generate better estimates for the time of secondary contact and ecotype divergence.

Importantly, secondary contact brought two lineages together that not only have evolved for many thousand generations in geographical isolation, but that are also phenotypically very different. They differ for example in their lateral plate phenotype, a nearly Mendelian trait encoded predominantly by the *Eda* gene on chromosome IV often under divergent selection between freshwater and marine environments[43]. In this case, one of the colonizing lineages is a freshwater lineage that has retained the otherwise marine, fully plated phenotype across its entire distribution range[30]. Fully plated East European freshwater stickleback may have been preadapted to environments with vertebrate predation regimes[44] such as large lakes like Lake Constance harboring a rich community of piscivorous fish[45]. Fully plated stickleback have an increased probability, relative to low-plated individuals, to escape and survive after capture by toothed predators such as piscivorous fish[46]. Plate morph is also known to be correlated with schooling behavior in stickleback, such that carriers of the fully plated allele show a higher tendency for schooling[47], which is common among marine stickleback and may be advantageous in the open waters of large lakes. Likewise, low-plated West European stickleback may have been relatively better pre-adapted to stream environments given the wide distribution of this lineage and its key phenotypes in rivers and streams of West Europe.

Introgression of West European alleles into streams in South-Eastern tributaries of Lake Constance may hence have been adaptive, as suggested by introgression into genomic regions that are enriched for QTL of lake-stream divergent traits such as jaw and dorsal spine length, lateral plate size and male coloration[3]. Whether lateral plate number and respective variation at the *Eda* locus is under selection in stream environments is unclear and

confounded by the origin of different stream populations: variation in genomic contributions from West East European lineages to stream ecotypes are correlated with lateral plate phenotypes (Fig. 1e), resulting in stream stickleback North and West of the lake being low plated and stream stickleback South-East of the lake being fully plated (Fig. 1e). Even though low-plated stickleback occur also in the South-East of Lake Constance, lake and stream ecotypes there are not divergent in this trait and stream ecotypes reduced lateral plate cover by reducing plate size instead of plate number[3]. A better understanding of ancestral phenotypes, of divergent selection between the lake and the stream habitat, the genomic architecture of traits and a higher genomic resolution will allow to identify the exact loci contributing to adaptation and reproductive isolation and to trace back their West, East European or recombined ancestry.

Secondary contact between old lineages can result in a complex mosaic of evolutionary outcomes. In classical work on contact zones between the West and East European species of firebelly toads (*Bombina bombina*, *B. variegata*), outcomes varied from persistence with leaky reproductive isolation in steep tension zones[48], through classical 'mosaic hybrid zone' patterns where each species occurs in habitat patches rich in the habitats they are best adapted to[49], to situations resembling a hybrid swarm in which allelic combinations are sorted between parental species habitats on very small spatial scale, reminiscent of ecological speciation from a hybrid population[50]. In Lake Constance stickleback, we document a similar continuum of outcomes where secondary contact and environmental adaptation interact in diverse ways. Outcomes range from partial collapse at the zone of contact of the old lineages at habitat boundaries (as between the GRA, NID, BOH stream population and the lake ecotype) to the assembly of alleles into new combinations in islands of genomic differentiation (S1 and S2 stream populations) that persist in sympatrically breeding ecotypes (S1 stream population versus lake ecotype).

It has long been known that gene flow in secondary contact can also lead to adaptive introgression of globally favorable alleles[51,52], which can lead to genomic islands of differentiation between lineages meeting in secondary contact[53,54]. Even though genomic islands among Lake Constance stickleback are derived from introgression between divergent lineages, the introgressed alleles are associated with divergent habitats and thus likely under divergent selection, rather than being globally favorable. In addition, in the case of Lake Constance stickleback, genomic islands did not only arise between formerly isolated populations near the zone of secondary contact, but repeatedly far outside the center of the hybrid zone where new stream ecotype populations evolved in situ. Introgression across the zone of contact has thus led to repeated ecological speciation in the geographical range (and genomic background) of one of the two old lineages. Our finding that genomic islands of sympatric ecotype divergence are derived from introgression of West European alleles suggests that introgression has likely facilitated or promoted early sympatric coexistence of ecotypes. The new stream ecotype populations carry a new combination of West and East European-derived alleles distinct from both original East and West European lineages (Fig. 6). Whether the new genomic combination resulted also in reproductive isolation of the new stream ecotype from the original West European lineage needs to be tested in the future.

Ecological speciation in stickleback from the South-East of Lake Constance bears resemblance to other cases of rapid speciation fueled by admixture variation. For example, in North American *Rhagoletis* flies a diapause allele introgressed from a distant Mexican population and facilitated colonization of new host plants and rapid ecological speciation in North America, thousands of kilometers away from the source of the alleles[5].

Admixture has also facilitated the repeated emergence of blue and red *Pundamilia* cichlid species in Lake Victoria, East Africa[4,55]. Introgression between divergent lineages and the resulting admixture variation may thus more generally be an important source of heritable variation for the rapid evolution of reproductively isolated species and for adaptive radiation[23–25,56,57].

## Methods

**Sample collection.** We used predominantly previously collected threespine stickleback populations listed in Supplementary Table 1 and collected three additional populations for this study (VAL, CHA, FRS11). Hand nets or electrofishing were used to capture stickleback at these sites, in accordance with scientific fisheries permits issued to members of the departmental federations for fisheries who executed the collection (department 84, Vaucluse; 89; Yonne, 03, Allier). Fish were euthanized in the field with an overdose of clove oil or MS-222 in accordance with the respective fisheries regulations.

**Mitochondrial DNA.** We obtained partial mitochondrial control region sequences for two populations. We extracted DNA from fin tissue using a Qiagen blood and tissue extraction kit, amplified the partial control region fragment using previously published primers (forward: 5′-CCTTTAGTCCTATAATGCATG-3′, reverse: 5′-CCGTAGCCCATTAGAAAGAA-3′)[26] and sequenced the fragment on an ABI 3130XL DNA Analyzer (Applied Biosystems)[22]. Accessions are given in Supplementary Table 1. We combined these sequences with previously published mitochondrial sequences from the Lake Constance region[12,21,22] and from populations across Europe[22,26,58], resulting in a combined dataset of 254 individuals from the Lake Constance catchment, each sequenced for 253 overlapping base pairs of the mitochondrial control region (Supplementary Table 1). Sequences were aligned manually in BIOEDIT v7.0.5.3[59], collapsed into identical haplotypes in MEGA X 10.0.5[60] and matched to European reference haplotypes Eu27, Eu36, Eu9, At1, Bs1, Sor1, Ner1, Ska1[26], CH01[22], and a new haplotype So17 identified in population DKM3. For these reference haplotypes, we concatenated partial mitochondrial control region and cytochrome b alignments of a total 1402 base pairs length, identified HKY + G + I as the best-fitting DNA substitution model based on BIC in MEGA and reconstructed the maximum-likelihood phylogeny under this model in MEGA, with support assessed from 1000 bootstrap replicates. We used the R-package ape 5.1[61] to visualize the phylogeny, using the following color code: yellow = So17, dark blue = Eu9, light blue = Eu36, green = CH01, orange = Eu27, black 1 = At1, 2 = Sor1, 3 = Ner1, 4 = Ska1, 5 = Bs1 (Fig. 1).

**Microsatellites.** We combined data for four microsatellite markers in common between two previous studies[12,21], featuring lake and stream populations from Lake Constance, populations from upper Danube, the upper Rhone catchment (Lake Geneva) and the River Rhine (Supplementary Table 1). We included 321 individuals without missing genotypes and used an admixture model in STRUCTURE 2.3.4[62] to infer population structuring with 50,000 burn-in steps followed by 300,000 steps in the MCMC chain. We ran STRUCTURE assuming 1–6 genetic clusters (K) with 10 replicates for each K. We identified the best number of genetic clusters supported by the data using the Evanno method[63].

**Morphological data.** We combined lateral plate morph data from lake and stream populations around Lake Constance, from the upper Rhone drainage (Lake Geneva), the Rhine and the upper Danube[3,12,21,22,64], in which individuals were scored as low, partially or fully plated morph, based on the presence of 0–3 lateral plates posterior the pelvic girdle (low plated), more than three later plates but with a gap of at least two plates to the caudal peduncle (partially plated) or the presence of a continuous series of lateral plates up to the caudal peduncle (fully plated)[21]. We embedded this lateral plate morph data in the context of historical phenotype distributions[22,30,65].

**Restriction-site-associated DNA.** We generated standard[3,66] restriction-site-associated (RAD) DNA sequence data for East (Vistula) and West (upper Rhone, Rhine) European stickleback populations (Supplementary Table 1) using the *Sbf*I-restriction enzyme, by single-end sequencing 100 bp up- and downstream of each restriction site on an Illumina HiSeq 2000. Accessions are given in Supplementary Table 1. We combined this data with previously published *Sbf*I data from Lake Constance for a Western tributary population and the adjacent lake site[33] and two South-Eastern tributary populations and their adjacent lake sites[3]. We additionally included overlapping *Pst*I-derived RAD sequencing data for further European populations and North American outgroups[27]. We also reanalyzed a *Nsi*I-derived RAD sequencing dataset[14], which, however, could not be merged with the *Sbf*I (+*Pst*I) data due to non-overlapping restriction sites. The *Nsi*I dataset includes three tributary populations from North and West of Lake Constance and adjacent lake sites, including two sites overlapping with the *Sbf*I dataset (NID, ROM, see Supplementary Table 1). Both *Sbf*I and *Nsi*I datasets were aligned to an improved version of the threespine stickleback reference genome[67,68] using BOWTIE 2 v2.0.0 with default parameters[69]. With PhiX reads available for *Sbf*I libraries only, we used

BASERECALIBRATOR from GATK v2.7[70] to calibrate base qualities of *SbfI*-reads[3]. We jointly called variants and genotypes in each *SbfI* and *NsiI* datasets using GATK's UNIFIEDGENOTYPER, using both SNP and insertions/deletions discovery mode, bases with minimum quality 20 and an assumed contamination rate of 3%[3]. From both datasets, we removed sites with depth higher than 1.5 times the interquartile range of the raw depth distribution, indels, sites on the sex chromosome XIX and sites with more than two alleles using BCFTOOLS v1.8[71] and VCFTOOLS v0.1.12b[72]. Then we considered genotypes with less than 10 reads depth as missing and removed sites with more than 50% missing data, resulting in 3,385,857 and 19,700,827 sites with 169,153 and 264,932 biallelic SNPs across 401 and 47 individuals for the *SbfI* and *NsiI* datasets, respectively.

For phylogenetic analysis, we used subsets of the *Sbf1* dataset with four to five individuals each for Lake Constance populations (ROM: 4, others: 5), three individuals each for European lineages and two each for North American outgroups (Fig. 1c), by selecting individuals with the lowest proportion of missing genotypes. In these phylogeny subsets, we removed monomorphic sites, sites with >25% missing data and sites with less than one homozygote for each allele and thinned the dataset to 100 kb distance between SNPs, resulting in 8205, 8144, 8144, 8173, and 8436 SNPs across 25, 24, 25, 25, and 39 individuals in datasets containing Lake Constance populations L2, ROM, NID, S1, and all combined, respectively. We used RAXML 8.0.26[73] to reconstruct the phylogenetic relationship amongst the individuals, using a generalized time-reversible (GTR) model with optimized substitution rates and a gamma model of rate heterogeneity. We further applied an ascertainment bias correction to account for the fact that we only used polymorphic SNP positions. Significance was assessed using 100 bootstrap replicates.

To infer population structuring, we used another *SbfI* subset containing all individuals from populations L1, L2, ROM, S1, S2, NID, PLS1, AGS1, and FRS4 (total *n* = 102). We reduced the dataset to 3030 biallelic SNPs with a minor allele frequency >5%, <5% missing genotypes and no linkage disequilibrium exceeding $r^2 = 0.95$ within 100 kbp windows. To avoid effects of allelic dropout due to a high proportion of PCR duplicates in some populations (L1, S1, NID, ROM), we randomly sampled one allele at each genotype to mimic the same level of allelic dropout for each sample, using a custom bash script (randdip2fakehomVCF.sh, v1.0)[3]. We then ran ADMIXTURE v1.3.0[74] for haploid data with otherwise default settings assuming 2–10 clusters (K), followed by cross-validation to identify the number of clusters with the lowest cross-validation error.

We used the same 102 individual subset of the *SbfI* dataset to test for excess allele sharing between Lake Constance stickleback populations and European sister lineages, resulting in a dataset of 23,277 SNPs. We added the alleles of three different outgroups, Japan Sea stickleback *Gasterosteus nipponicus*, Black-spotted stickleback *G. wheatlandi* and Ninespine stickleback *Pungitius pungitius* to this dataset, inferred from previously published whole-genome data[75] (Sequence Read Archive accessions DRR032274, DRR013347, and DRR013346). We aligned all three outgroups to the improved threespine stickleback reference assembly[68] using STAMPY v1.0.22[76] with default settings except for adding the option '--substitutionrate' to add the expected divergence to the reference genome as estimated by Yoshida et al.[75]. We used FREEBAYES v1.1.0-3-g961e5f3-dirty[77] to call outgroup genotypes for the 23,277 SNP positions in our *SbfI* dataset for genotypes with minimal depth 6 and phred-scaled site and genotype quality 20, then merged variants with the *SbfI* dataset and removed any multiallelic sites or indels using BCFTOOLS. We then computed the D-statistic using ADMIXTOOLS v4.1[78] for different quartets featuring each of the three outgroup taxa (P4/out in Supplementary Fig. 3 phylogenetic tree). When testing for excess allele sharing between all Lake Constance populations and West European lineages, we used PLS1 (P1) paired with all Lake Constance populations (P2) as ingroup taxa and the two West European populations FRS4 and AGS1 as potential source of introgression (P3, Supplementary Fig. 1). When testing for excess allele sharing of one Lake Constance ecotype over the other with West European lineages, we used one lake (P1: L1, L2, ROM) and one stream (P2: S1, S2, NID) population each as ingroup sisters and the three European populations AGS1, FRS4, and PLS1 as potential sources of introgression (P3, Fig. 2a, Supplementary Fig. 3).

We obtained rough estimates for admixture proportions of Lake Constance populations derived from West and East European lineages by computing the hybrid index with the R-package INTROGRESS v1.2.3[79]. For this, we identified a subset of SNPs in the *SbfI* dataset which are divergently fixed between the populations PLS1 and the sum of the two populations AGS1 and FRS4, as assessed from a minimum of three genotypes for each group, resulting in 299 SNPs. Genotypes at these SNPs were used as input to estimate hybrid index with INTROGRESS using default settings. We caution that the low number of individuals in PLS1 may lead to the erroneous inclusion of ancestrally polymorphic SNPs, which should lead to an overestimation of West European admixture proportions, in line with higher estimates obtained with this method than with demographic modeling (see Results).

For demographic modeling, we converted subsets of both *Sbf1* and *Nsi1* datasets into multidimensional site-frequency spectra (SFS). The *Sbf1* data subsets included one lake population (L2), one South-Eastern tributary stream (S2) population, one Western tributary stream (NID) population, the East European lineage (PLS1), and two West European lineages from the Rhine (AGS1) and the upper Rhone (FRS4), depending on the hierarchical level tested (see Results, Fig. 3). The *Nsi1* dataset featured one lake (ROM) and three Northern and Western tributary stream

populations (BOH, GRA, NID), with NID overlapping between *NsiI* and *SbfI* datasets. We excluded individuals with a strong allele imbalance in heterozygous genotypes, i.e., a minor/major read number imbalance strongly deviating from 1:1 (Supplementary Fig. 12), indicative of PCR errors or other artifacts, which may appear as singletons[80] thus biasing the site-frequency spectrum. This left us with the following number of suitable individuals for demographic modeling: L2: 10, S2: 8, PLS1: 4, AGS1: 3, FRS4: 6, ROM: 7, BOH: 10, GRA: 9, NID (*SbfI*: 8, *NsiI*: 15). In each population, we identified loci out of Hardy–Weinberg equilibrium using VCFTOOLS and discarded sites with evidence of heterozygote excess plus/minus a buffer of 100 bp left and right of each site from the dataset in order to eliminate putative duplicated regions in the genome and removed the sum of all sites. Then, we removed sites in low recombination rate regions of the genome (<1.5 cM Mbp⁻¹) to avoid biases due to linked selection[32]. We estimated local recombination rates based on the FTC cross recombination map[68] by cubic spline smoothing of the genetic on the physical map for each chromosome with a spline parameter of 0.7 and calculating first derivatives (= recombination rates) for positions of interest[3]. Next, we randomly subsampled a fixed number of genotypes from each population (L2: 6, S2: 6, PLS1: 4, AGS1: 3, FRS4: 4, ROM: 7, BOH: 9, GRA: 9, NID: 9/*NsiI* and 6/*SbfI*) to a dataset without missing data, discarding sites with too few genotypes, with a custom python script (sampleKgenotypesPerPop.py, v1.1). We further reduced the SNP portion of the datasets to unlinked SNPs by removing sites with $r^2 > 0.95$ within 200 bp distance. After this, we added again *N* monomorphic sites proportional to the retained number of SNPs post linkage-pruning by randomizing the order of monomorphic sites and retaining the first *N* sites. Finally, we converted the resulting linkage-pruned datasets without missing data into multidimensional SFS using a custom python script (vcf2sfs.py, v1.1), folded them with another custom python script (foldSFS.py, v1.0) and converted them to joint 2D-SFS with a custom R script (SFStools.R, v1.1). We also generated 100 non-parametric block-bootstrap replicates of the observed multidimensional SFS by randomly resampling blocks of 10,000 sequenced sites adjacent to each other on a chromosome to the observed number of such blocks as implemented in vcf2sfs.py.

**Demographic modeling**. We used fastsimcoal2 v2.6[81] and a hierarchical modeling approach to reconstruct the demographic history of Lake Constance stickleback by comparing the fit of different demographic models to observed SFS and to estimate parameters for the best-fitting models. First, we optimized three population models (Fig. 3a, Supplementary Fig. 7) on the observed, *SbfI* derived, folded 3D-SFS with the putative sister lineages to Lake Constance stickleback from the Vistula (PLS1), Rhine (AGS1), and upper Rhone (FRS4). For each model, we maximized the likelihood from 100 random starting parameter combinations and minimal 10 to maximal 50 ECM cycles with a stopping criterion of 0.001[81]. The expected minor allele (folded) SFS for each model and parameter combination was approximated with 100,000 coalescent simulations. We used a mutation rate of 1.7E−8 in all simulations[82]. The likelihood and parameter estimates of each model were obtained from the run with the highest likelihood among 100 optimizations. To identify the best-fitting model, the likelihoods of models were compared using the Akaike information criterion (AIC)[81]. We estimated confidence intervals for parameters of the best model from non-parametric block-bootstrap replicates of the observed 3D-SFS: we performed 10 parameter optimizations with each bootstrap replicate, starting from the best parameters inferred from the observed data, and the parameter estimates from the run with the highest likelihood for each bootstrap sample was used to compute 95% confidence intervals using empirical percentiles. Furthermore, we assessed whether other models than the best model explained the data similarly well or not by computing likelihood distributions from 100 SFS simulated under each model given the maximum-likelihood parameters inferred from the observed data and comparing the overlap of these likelihood distributions (Supplementary Fig. 8)[4].

After identifying the best model (model 3a, see Fig. 3a and Results section), we used the 95% confidence intervals for divergence times between West and East European lineages, upper Rhone and Rhine lineages and the population sizes of the three populations as constraints for parameter searches in more complex models. In a next step, we added single Lake Constance populations (S2, NID, L2) to the sister lineage trios and compared four population models (Fig. 3b, c) optimized on *SbfI* derived, observed, folded, joint 2D-SFS, as outlined above. Finally, we added one lake and one stream population each to the sister lineage trios and compared five population models representing different modes of ecotype divergence in Lake Constance (Fig. 3d). We optimized these models on *SbfI* derived, observed, folded joint 2D-SFS for lake-stream pairs L2 vs. S2, L2 vs. NID and on *NsiI* derived, observed, folded joint 2D-SFS for lake-stream pairs ROM vs. NID, ROM vs. BOH and ROM vs. GRA. We modeled the three sister lineages as unsampled populations for optimizations on the *NsiI* dataset and fixed their population parameters either to the maximum-likelihood parameters estimated in model 3a (Supplementary Fig. 6) or constrained the search range as outlined above. Both approaches led to the same models emerging as best supported models (Table 1, Supplementary Table 2). We repeated the same optimizations assuming constrained-parameter or fixed-parameter unsampled sister lineages with the *SbfI* dataset, which also led to the same models emerging as best supported models (Table 1, Supplementary Table 2) even though the power to distinguish the models significantly decreased (Supplementary Fig. 8). In model optimizations on *SbfI*-derived data including the

population NID, we ignored singletons in the likelihood computation, the former to avoid possible remaining PCR artifacts in this population.

**Population differentiation**. We used a subset of the *Sbf1* dataset to test for introgression of West European lineages into Lake Constance. This dataset contained one lake (L1) and two South-Eastern tributary populations (S2, S1), one East European (PLS1) and two West European populations (AGS1, FRS4). We filtered the *Sbf1*-data subset for each pairwise comparison to include only biallelic SNPs with at least three sequenced genotypes per population using BCFTOOLS. We estimated pairwise F-statistics for each SNP by performing locus-by-locus AMOVAs in ARLEQUIN v3.5.23. SNP-level F-statistics were averaged over non-overlapping windows containing at least 2500 sequenced bases without splitting RAD loci or over sliding windows of 1 Mbp size with 200 kbp step size. Non-overlapping windows were on average 344 kb (73–1564 kb) wide. We computed an average recombination rate for each window by computing the mean over 10 recombination rate estimates at equally spaced positions across each non-overlapping window. We computed weighted average F-statistics by averaging variance components[83] and calculating the ratio of averages[84]. We used permutation tests to assess, whether differentiation in windows overlapping with genomic islands of differentiation persisting among sympatric populations identified in an earlier study[3] differed from the genome-wide distribution of differentiation. For this, we permuted the position of genomic islands 10,000 times across the genome, generated a null distribution for the mean and computed empirical quantiles for the observed mean differentiation in genomic islands.

We formally tested whether parallel genomic islands of differentiation between Lake Constance ecotypes identified previously[3] rather represent genomic regions resisting gene flow between ecotypes or regions that diverged due to background selection in the absence of gene flow, using a similar subset of the *Sbf1* dataset as above with one South European (SOR) and one West European population (CHA) added, as well as the full *NsiI* dataset. In non-overlapping windows containing at least 2500 SbfI-RAD sequenced sites, we computed differentiation (weighted average $F_{ST}$) for all pairwise comparisons as outlined above. In addition, we estimated nucleotide diversity ($\pi$) within population and pairwise absolute divergence ($d_{XY}$) for the same non-overlapping windows. For the latter two statistics, we subsampled SbfI- or NsiI-data subsets of population pairs to N genotypes per population corresponding to 75% of a population's individuals in order to get a dataset without missing data, discarding sites with too few genotypes, with the custom script sampleKgenotypesPerPop.py. We summarized the subsampled VCF files into 2D-SFS for each non-overlapping window with the custom script vcf2sfs.py and computed both $\pi$ and $d_{XY}$ with custom scripts (wsfs_dxy.py, v1.0; wsfs_pi.R, v1.1), respectively. Pearson's correlation coefficient ($r$) between statistics was computed in R for windows without missing values, for pairwise statistics as $F_{ST}$ or $d_{XY}$ using statistics from non-overlapping population pairs to avoid autocorrelation. All data analysis was performed on the servers of the Genetic Diversity Centre (GDC), the Ubelix computer cluster, University of Bern and the Euler cluster, ETH Zurich, Switzerland.

**Reporting summary**. Further information on research design is available in the Nature Research Reporting Summary linked to this article.

## Data availability

All morphological and genetic data used in this study is available from previously published datasets or accessions detailed in Supplementary Table 1. These data are also available from the corresponding author on reasonable request. Sequence data has been deposited on GenBank under accessions MN082769–MN082781 and on Sequence Read Archive under accessions SRR9317386–SRR9317452, SRR9335375–SRR9335380 and SRA-BioProject accession PRJNA549360. The source data underlying Figs. 1–6, Table 1, Supplementary Figs. 1–11 and Supplementary Table 2 are provided as a Source Data file.

## Code availability

All custom scripts mentioned in the Methods section have been deposited on GitHub and are accessible under https://github.com/marqueda

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

## Acknowledgements

We thank Henry Persat for facilitation of field work in France, Yann Monnier, Claude Chadefaux, Mickael Lelièvre, Jean Louis Clere for collecting stickleback in France and Rafał Bernas for collecting stickleback in Poland, Joana Meier, Carmela Doenz and students of the Seehausen lab for assistance in the field, Salome Mwaiko for assistance in the lab, Aria Minder and Stefan Zoller from the Genetic Diversity Center, ETH Zurich/ Eawag for bioinformatics support and all members of the Fish Ecology and Evolution Department at Eawag and the Computational and Molecular Population Genetics lab for discussion. This research was supported by the Swiss National Science Foundation (SNF) grants PDFMP3_134657 and 31003A_163338 to O.S. and L.E. V.S. is funded by EU H2020 program (Marie Skłodowska-Curie grant 799729).

## Author contributions

O.S., L.E., K.L., and D.A.M. conceived the study. K.L., D.A.M., and O.S. acquired samples. K.L. and D.A.M. generated and compiled morphological and genetic data. K.L. conducted morphological, microsatellite, and mitochondrial analyses, D.A.M. genomic analyses and demographic modeling, with input from V.S. L.E. and D.A.M. created code. D.A.M. and K.L. created figures. D.A.M. and K.L. wrote the paper, with input from O.S., L.E., and V.S. O.S., L.E., and V.S. acquired funding.

## Additional information

**Competing interests:** The authors declare no competing interests.

