## [Peer Review File · Nature Communications]

Reviewers' Comments:

Reviewer #1:

Remarks to the Author:

The manuscript presented here focuses on the very recent speciation/invasion event of Lake Constance by sticklebacks using a combination of morphological /mtDNA/Microsats/RAD approach. The claim is that secondary contact and hybridization resulted in rapid adaptation that allowed for speciation and invasion. Morphological, phylogeographic and nuclear genomic arguments are provided by testing alternative hypotheses.

This paper is original, bold, and extremely interesting.

The paper is very clearly written, provides a good background of the issue and represents, very fairly (in my opinion) what alternative scenarios have been proposed in the past. Such alternative scenarios were scrutinized critically and subjected to the same model testing as the data presented here and found to have a lower fit to the model than proposed in the past.

This paper provides a new way of looking at speciation and invasion and offers original clues as to the potential mechanisms of rapid adaptation, a central issue that has vexed scientists for a long time. In my mind this paper is nothing short than brilliant, as it finally provides some true originality to the field.

I have only one very minor comment on the paper.

Figure 2 is the crux of the paper, yet I find it very poorly designed. I wonder of the use of colors would make it more palatable. As it is, it is difficult to distinguish between models unless the figure is enlarged quite a bit.

Figure 3 could use the same patterns of color, but it is not as crucial as Figure 2.

Reviewer #2:

Remarks to the Author:

The authors present an analysis of population structure in Lake Constance stickleback and stream populations across Europe based mostly on published RAD-seq datasets. Through genome scans for F_{st} and demographic analysis, they argue that in one small area of the lake where 'lake' and 'stream' ecotypes coexist, the 'stream' ecotypes share more differentiated regions with Rhine populations than 'lake' ecotypes. They conclude a 'hybrid swarm' scenario of speciation in this case study.

This is surely a large sampling effort that will increase knowledge of this lake stickleback system, however, I have major reservations with the analyses, framing, and conclusions drawn from these data. This study would be more suitable for a discipline specific journal, such as Molecular Ecology or Evolutionary Ecology Research.

- 1) The 'genomic islands of speciation' concept has now been discredited numerous times (e.g. Cruikshank and Hahn) and the authors do nothing to address these concerns, such as presenting additional population genetics statistics needed.
- 2) They have shown variable population structure in one area of the lake, but no data is presented to link this variation to ecologically-relevant traits. Simply performing an F_{st} -scan is not sufficient.
- 3) This manuscript is written almost entirely about the Lake Constance stickleback. Very little effort is given to discuss other systems or connect these results in a broader context.

4) The data presented has already been analyzed using similar analyses in previous publications. The only new data appears to be the addition of one more outgroup.

Comments -

1) The introduction is highly stickleback specific – in fact, nearly all citations are to this European stickleback system.

2) Overall, there is no evidence that the different sources of genetic variation actually provide important genetic variation to speciation as the authors claim.

3) Many results are a laundry list of phylogeographic clusters – this is not relevant to the broad audience of Nat Comm.

4) The entire study appears to be a reanalysis of published SNP data with the addition of one more outgroup.

5) The authors present an analysis of genome-wide F_{st} . They have not adequately tested for the presence of introgression (e.g. D-statistics), absolute divergence (D_{xy}), selection, or recombination rate variance from existing stickleback linkage maps.

6) Claims of admixture variation brought in by secondary contact were of crucial importance (line 91) or fueled rapid ecological speciation (lines 33,260,674) does not seem fully justified by the data presented. The evidence presented makes this hypothesis that admixture facilitated rapid ecological speciation definitely worth further investigation, but I don't think that this study actually shows that secondary contact/admixture did play a crucial role as the language used in the paper seems to suggest. There is no real story presented of how that variance might have driven ecological differentiation. I don't think that overlap with genomic islands is enough for such claims as regions responsible for reproductive isolation/under divergent selection are not only explanation for peaks of differentiation in genome, and no additional evidence here that suggest these regions are/were important to ecological divergence.

7) When describing the models tested, there is a lot of time spent on the primary divergence ones, but not a lot explaining others or the reasoning behind some of the variation in the models shown in Fig 2 (particularly the hybrid speciation/secondary contact scenarios that are most supported in the end). I would like to see at least equal explanation of these later two and the rationale between the variation tested as given to the primary divergence/ecological vicariance models. I think this might also help clarify your definition of hybrid speciation.

8) Regarding lines 216-218: Is it really a hybrid speciation scenario if introgression is dated being later than the initial divergence of the lake/stream ecotypes or is it more like secondary gene flow after initial divergence has begun? It may help to clarify what the hybrid speciation scenario you are testing is. There is also little discussion of why different scenarios supported by different datasets and it seems like the secondary contact scenario was chosen over the other for the rest of the paper. I would like to see the logic behind this choice flushed out a little more.

9) On lines 236-238 predict that genomic islands between stream and lake ecotypes would show differential signatures of introgression if introgression of Western alleles facilitated ecological speciation. It could also be possible that introgression occurred in the ancestor of the ecotypes and then introgressed alleles were differentially sorted between the two, so the signature of introgression

could exist in both stream and lake ecotypes.

10) A little more background context for the system, such as a sentence or two describing the two ectomorphs and their differences would be nice for the broad audience of Nature Communications that might not know much about sticklebacks.

11) Not quite clear why re-analyzing microsatellite data was necessary when you can run STRUCTURE with genome-wide SNP data. Maybe crucial populations missing from SNP dataset?

12) Possible to make a phylogenetic network from SNP data? This may complement findings of secondary contact since bifurcating trees would not be the best way to represent an evolutionary history involving admixture as suggested. Why aren't the N/W populations in phylogeny of Fig 1c? What does the current phylogeny in the paper really add that wasn't mentioned in the introduction that someone else found previously about phylogenetic placement of lake populations inside stream (line 72)?

13) In Fig 2 and Fig 3. The order of panels doesn't follow the order that they are discussed in the paper. I would suggest making the ordering consistent between the two for ease of reading.

14) What do the dashed lines in Fig 5 represent? Average F_{st} ? This information should be added to the caption for the figure.

Reviewer #3:

Remarks to the Author:

The history of stickleback ecotypes in the Lake Constance basin is the subject of an ongoing debate, in which the current authors hold a view differing from that of Roesti, Berner, and colleagues. The paper by Roesti et al (2015) was also published in Nat Comms, laying out the scenario that the basin was first colonized by stream-adapted fish, and the lake population is derived from those ancestors. The authors here do a fair job of presenting that alternative view in the Introduction and addressing it in the work here. They take multiple perspectives (genetic datasets, demographic models) to address the issues. However, there are a few things that could be added to strengthen the manuscript.

- mtDNA haplotype and microsat data show a strong signal of association with eastern European/Baltic origins for the Lake Constance populations, with input also from other sources. This aligns with phenotypic data on plate morphs, in which the lake populations are dominated by high-plated fish, as are the freshwater populations in eastern Europe.

- Newly generated RAD sequencing data for this study also support this scenario of lake and southeast stream populations grouping with eastern European source populations, but this is only tested on 2 populations very close to each other in the southeast portion of the lake. It is extremely unfortunate that the RAD data from the Roesti et al work is not compatible with this group of authors!

- Even though the RAD data are distinct between the datasets, different sets of demographic scenarios should not be tested on each dataset, or else there is not a direct comparison of competing hypotheses. The hybrid speciation and secondary contact models are not well-explained in the text, but HybSpe-M-c seems to have a bottleneck that is not represented in the R tests. These scenarios must be consistent across datasets — if they are and I misunderstand the scenarios, they should be better described.

- The evidence for adaptive introgression based on F_{st} is an interesting result. It would be instructive to show the regions of reduced F_{st} between western and stream populations along the genome, and compare to the stream-lake and stream-eastern comparisons. With the focus on lateral plate phenotype, the authors should look at patterns of differentiation around the *Eda* locus in particular.
- It is always worth considering multiple values of k in Structure analyses, not just the best-supported one. What do larger values of k show with the microsat data? Perhaps a supplemental figure could show this.
- The abbreviations used for demographic models (e.g. lines 174-185) are a bit unclear at first.
- Fig 1: Are there multiple haplotypes in the potential source populations that are not found in the Lake Constance basin? The filled circles in part (a), set next to the pie charts in part (b), suggest no, but what is the haplotype diversity in the source populations? And how much haplotype variation is shared among source populations? Also in 1c, the lowest C1 label on the tree should be the same font size as the other C1 labels.
- Table 1: Can references be given directly in the table, rather than letters that are then linked in the caption? Check the reference for NID.

Detailed responses to referee comments

Reviewer #1

The manuscript presented here focuses on the very recent speciation/invasion event of Lake Constance by sticklebacks using a combination of morphological / mtDNA / Microsats / RAD approach. The claim is that secondary contact and hybridization resulted in rapid adaptation that allowed for speciation and invasion. Morphological, phylogeographic and nuclear genomic arguments are provided by testing alternative hypotheses.

This paper is original, bold, and extremely interesting. The paper is very clearly written, provides a good background of the issue and represents, very fairly (in my opinion) what alternative scenarios have been proposed in the past. Such alternative scenarios were scrutinized critically and subjected to the same model testing as the data presented here and found to have a lower fit to the model than proposed in the past.

This paper provides a new way of looking at speciation and invasion and offers original clues as to the potential mechanisms of rapid adaptation, a central issue that has vexed scientists for a long time. In my mind this paper is nothing short than brilliant, as it finally provides some true originality to the field.

I have only one very minor comment on the paper. Figure 2 is the crux of the paper, yet I find it very poorly designed. I wonder of the use of colors would make it more palatable. As it is, it is difficult to distinguish between models unless the figure is enlarged quite a bit. Figure 3 could use the same patterns of color, but it is not as crucial as Figure 2.

We thank reviewer #1 for the positive assessment of our manuscript. We have redesigned all figures, including the figures presenting the models and parameter estimates (now Figs. 3-4, S6-S7) along the lines suggested.

Reviewer #2

The authors present an analysis of population structure in Lake Constance stickleback and stream populations across Europe based mostly on published RAD-seq datasets. Through genome scans for F_{st} and demographic analysis, they argue that in one small area of the lake where 'lake' and 'stream' ecotypes coexist, the 'stream' ecotypes share more differentiated regions with Rhine populations than

'lake' ecotypes. They conclude a 'hybrid swarm' scenario of speciation in this case study.

This is surely a large sampling effort that will increase knowledge of this lake stickleback system, however, I have major reservations with the analyses, framing, and conclusions drawn from these data. This study would be more suitable for a discipline specific journal, such as *Molecular Ecology* or *Evolutionary Ecology Research*.

We thank reviewer #2 for the constructive feedback on our manuscript.

R2.1 1) The 'genomic islands of speciation' concept has now been discredited numerous times (e.g. Cruikshank and Hahn) and the authors do nothing to address these concerns, such as presenting additional population genetics statistics needed.

We have added two paragraphs to the results section (lines 333-361) as well as a supplementary figures (Fig. S9-S11) presenting additional population genetic statistics and discussing alternative hypotheses for the origin of genomic islands reported earlier (Marques *et al.* 2016). The alternative hypotheses being (a) genomic islands arising due to linked selection in the absence of gene flow or (b) due to reduced gene flow following from divergent selection or reproductive isolation. We have now for the first time compared allopatric differentiation landscapes between populations from different European watersheds (no gene flow scenario) with the sympatric / parapatric ecotype differentiation landscapes in Lake Constance (opportunity for gene flow). We found that F_{ST} -differentiation landscapes among allopatric populations are negatively correlated with nucleotide diversity and positively with d_{XY} , but not correlated with recombination rate (Fig. S9), supporting the linked selection hypothesis (in this case: local adaptation leading to reduced diversity and high d_{XY} , rather than background selection leading to reduced diversity / d_{XY} and high F_{ST} in low recombination regions) for genomic differentiation in the absence of gene flow in allopatry. However, sympatric / parapatric and allopatric F_{ST} -differentiation landscapes are not correlated with each other, contrary to expectations of the linked selection hypothesis. Furthermore, sympatric / parapatric F_{ST} -differentiation landscapes are neither correlated with diversity or d_{XY} (Fig. S9), nor do genomic islands fall into regions of strongly reduced diversity (Marques *et al.* 2016), as would be expected under the linked selection hypothesis. Instead, sympatric / parapatric ecotype differentiation across the genome is correlated between replicate streams and thus associated with habitat differences. Ecotype differentiation is also negatively correlated with recombination rate. Our results thus suggest that either divergent selection between habitats or reproductive isolation maintain allele frequency differences associated with ecotypes against gene flow in sympatry / parapatry, and they do not indicate a role of background selection.

R2.2 2) They have shown variable population structure in one area of the lake, but no data is presented to link this variation to ecologically-relevant traits. Simply performing an Fst-scan is not sufficient.

Establishing a link to ecologically relevant traits has been the focus of our previous manuscript, where we found an enrichment of QTL controlling divergent traits between ecotypes in genomic islands of differentiation (Marques *et al.* 2016). We have revised our manuscript and now mention the QTL-enrichment explicitly (lines 64-65, 332, 481-483) and also discuss the role of ancestral lateral plate phenotypes in more detail, which is one well-known ecologically relevant trait (lines 466-491). We fully agree with reviewer #2 that additional research is needed to connect causal genotypes with phenotypes and reproductive isolation in this system and discuss future challenges arising from our findings (lines 491-495).

R2.3 3) This manuscript is written almost entirely about the Lake Constance stickleback. Very little effort is given to discuss other systems or connect these results in a broader context.

We have revised our discussion to focus on the unexpected outcome of secondary contact between lineages – triggering repeated ecological speciation. In that context, we are now discussing other systems, e.g. the diverse *Bombina* hybrid zones, *Rhagoletis* flies, *Pundamilia* cichlids, and the broader idea of speciation through re-assembly of old alleles into new combinations.

R2.4 4) The data presented has already been analyzed using similar analyses in previous publications. The only new data appears to be the addition of one more outgroup.

We respectfully disagree with the reviewer. In our biogeographic reconstruction, we show that reconstructing the demographic history and the origin of alleles is only possible because we combine previously published data into novel, previously unpublished RAD-sequencing data from West and East European sister lineages (populations AGS1, FRS4, PLS1). This new data also allowed us to detect that genomic islands are largely introgression-derived.

Regarding analyses, our manuscript is the first to apply an exhaustive demographic modelling analysis to any stickleback speciation system, while in the only previous attempt a single model was applied and alternatives were not even tested (Roesti *et al.* 2015).

Comments -

R2.5 1) The introduction is highly stickleback specific – in fact, nearly all citations are to this European stickleback system.

We have revised the introduction accordingly. The focus lies on the origin of alleles involved in ecological speciation and we cite now a few examples for which such an origin has successfully been established. Even though an in-depth introduction in the Lake Constance stickleback system is necessary in our view, we have now strongly condensed this part.

R2.6 2) Overall, there is no evidence that the different sources of genetic variation actually provide important genetic variation to speciation as the authors claim.

We respectfully disagree: the genomic islands of differentiation between ecotypes – for which we demonstrate their different source origins – are enriched with QTL for traits diverging between ecotypes (Marques *et al.* 2016), i.e. they contain loci coding for phenotypic differences between ecotypes. Furthermore, these genomic differences persist despite sympatric breeding of the ecotypes. Therefore, the introgressed and divergently sorted regions bear relevance for ecologically relevant traits and coexistence in sympatry, and are therein relevant for ecological speciation.

R2.7 Many results are a laundry list of phylogeographic clusters – this is not relevant to the broad audience of Nat Comm.

We have completely removed the ‘laundry list’ and rewritten the results section. The revised results now only briefly present phylogeographic patterns important for the main story regarding geographical sources of alleles.

R2.8 4) The entire study appears to be a reanalysis of published SNP data with the addition of one more outgroup.

Please see our response to the earlier comment R2.4.

R2.9 5) The authors present an analysis of genome-wide F_{ST} . They have not adequately tested for the presence of introgression (e.g. D-statistics), absolute divergence (D_{xy}), selection, or recombination rate variance from existing stickleback linkage maps.

We have added D-statistics and Hybrid index estimations to adequately test for and quantify admixture – confirming our hypothesis of secondary contact / admixture. In addition, we have analyzed genome-wide correlation of F_{ST} with nucleotide diversity, d_{XY} and recombination rate variation to assess whether background selection could explain genomic islands (see our response to R2.1). Also, note that

we had addressed signatures of selection in genomic islands as far as feasible with RAD-sequencing data in our previous paper (Marques *et al.* 2016).

R2.10 6) Claims of admixture variation brought in by secondary contact were of crucial importance (line 91) or fueled rapid ecological speciation (lines 33,260,674) does not seem fully justified by the data presented. The evidence presented makes this hypothesis that admixture facilitated rapid ecological speciation definitely worth further investigation, but I don't think that this study actually shows that secondary contact/admixture did play a crucial role as the language used in the paper seems to suggest. There is no real story presented of how that variance might have driven ecological differentiation. I don't think that overlap with genomic islands is enough for such claims as regions responsible for reproductive isolation/under divergent selection are not only explanation for peaks of differentiation in genome, and no additional evidence here that suggest these regions are/were important to ecological divergence.

We have added additional results on alternative explanations for differentiation peaks (see our response to R2.1). Furthermore, we have expanded our 'story' in the discussion on how introgressed variation may have facilitated ecological differentiation, based on the phenotypic traits divergent between ecotypes and between the different sources (e.g. lateral plates) and based on QTL enrichment of divergent traits in genomic islands (lines 466-491).

R2.11 7) When describing the models tested, there is a lot of time spent on the primary divergence ones, but not a lot explaining others or the reasoning behind some of the variation in the models shown in Fig 2 (particularly the hybrid speciation/secondary contact scenarios that are most supported in the end). I would like to see at least equal explanation of these later two and the rationale between the variation tested as given to the primary divergence/ecological vicariance models. I think this might also help clarify your definition of hybrid speciation.

8) Regarding lines 216-218: Is it really a hybrid speciation scenario if introgression is dated being later than the initial divergence of the lake/stream ecotypes or is it more like secondary gene flow after initial divergence has begun? It may help to clarify what the hybrid speciation scenario you are testing is. There is also little discussion of why different scenarios supported by different datasets and it seems like the secondary contact scenario was chosen over the other for the rest of the paper. I would like to see the logic behind this choice flushed out a little more.

We agree with the criticism on naming this model 'hybrid speciation', in which indeed we would predict introgression to either pre-date or to happen simultaneously with ecotype divergence. We have accordingly renamed the scenario 'hybrid origin', highlighting the feature that one of the ecotypes is modelled as being of admixed origin, as opposed to different origins of each ecotype in a secondary contact

scenario. We define this scenario as follows: “...in which one of the ecotypes has recently evolved through hybridization between divergent West and East European lineages or sorting of admixture variation following introgression from one divergent lineage into the other lineage” (L95-97). Renaming the model also ensures that the demographic model – a neutral model optimized on predominantly neutrally evolving sites in the genome (high-recombination rate region RAD-SNPs) – does not insinuate the inclusion of adaptive processes into the modelling, even though adaptation to different habitats and thereby differential introgression or differential sorting of alleles was likely involved in ecotype divergence (see our reply to the next comment).

We have revised our explanation of each scenario, now in the second last introduction paragraph (lines 89-97, Fig. 3), and discuss why the hybrid origin models better capture the observed data than secondary contact and primary divergence models (lines 275-283).

R2.13 9) On lines 236-238 predict that genomic islands between stream and lake ecotypes would show differential signatures of introgression if introgression of Western alleles facilitated ecological speciation. It could also be possible that introgression occurred in the ancestor of the ecotypes and then introgressed alleles were differentially sorted between the two, so the signature of introgression could exist in both stream and lake ecotypes.

We agree that both scenarios – differential introgression or differential sorting of alleles after introgression – will lead to excess allele sharing between stream ecotype and West European populations.

With our demographic models and summary statistics (D-stats, hybrid index), we cannot distinguish the two, because the former assume neutrality and are based mostly on neutrally evolving sites in the genome (high-recombination rate region RAD-SNPs) and the latter are able to detect certain patterns (excess allele sharing) that might arise from either process of introgression and/or selective sorting, but the methods cannot distinguish between the two processes underlying this pattern. The patterns we detect with these methods are that both, the lake ecotype and the stream ecotype in South-Eastern Lake Constance show signatures of introgression (Fig. 2b), but the stream ecotype shows stronger allele sharing with West European populations (Fig. 2a, 4b). We have accordingly changed the wording in the results section to reflect the pattern (excess allele sharing) rather than the process (differential introgression vs. differential sorting after non-adaptive introgression) in the paragraphs dealing with D-stats, hybrid index and demographic modelling. An exception is the section where we describe introgression proportions for the demographic models where indeed directional introgression was modelled (a neutral

process). Inferred directional introgression may capture either differential introgression or differential sorting after non-selective introgression in nature.

Both processes will generate admixture variation and we thus think that both are relevant to understanding the role of admixture variation in ecological speciation. Our analysis of genomic islands underlines this relevance: the genomic islands we study are regions resisting gene flow where ecotypes coexist and showing habitat-allele associations (Marques *et al.* 2016). Their persistence and habitat-association means that either differential introgression led to different admixture proportions that are now maintained in sympatry, or that previously introgressed alleles have been sorted and are now maintained in sympatry, with both sorting and maintenance due to divergent selection or assortative mating.

R2.14 10) A little more background context for the system, such as a sentence or two describing the two ecomorphs and their differences would be nice for the broad audience of Nature Communications that might not know much about sticklebacks.

We added two sentences to the introduction as suggested.

R2.15 11) Not quite clear why re-analyzing microsatellite data was necessary when you can run STRUCTURE with genome-wide SNP data. Maybe crucial populations missing from SNP dataset?

Good point, we now present clustering analyses for both genome-wide SNP data and microsat data in the supplementary figures Fig. S4-S5. In the microsat data, we can combine both sister lineages and all Lake Constance lake and stream populations, something that is not possible for either the *SbfI*- or the *NsiI*-RAD-sequencing dataset.

R2.16 12) Possible to make a phylogenetic network from SNP data? This may complement findings of secondary contact since bifurcating trees would not be the best way to represent an evolutionary history involving admixture as suggested. Why aren't the N/W populations in phylogeny of Fig 1c? What does the current phylogeny in the paper really add that wasn't mentioned in the introduction that someone else found previously about phylogenetic placement of lake populations inside stream (line 72)?

We have expanded our phylogenetic analysis by analyzing several subsets of the data, which now show that different Lake Constance populations cluster differently in a phylogenetic tree. We fully agree that a bifurcating tree is not an ideal representation of the data and therefore place the emphasis on our demographic modelling results. We think the expansion of our demographic analyses is more useful here than adding a phylogenetic network.

R2.17 13) In Fig 2 and Fig 3. The order of panels doesn't follow the order that they are discussed in the paper. I would suggest making the ordering consistent between the two for ease of reading.

Changed as suggested with the redesigned figures and rewritten results section.

R2.18 14) What do the dashed lines in Fig 5 represent? Average F_{st} ? This information should be added to the caption for the figure.

Changed and now included in the figure caption.

Reviewer #3

The history of stickleback ecotypes in the Lake Constance basin is the subject of an ongoing debate, in which the current authors hold a view differing from that of Roesti, Berner, and colleagues. The paper by Roesti et al (2015) was also published in Nat Comms, laying out the scenario that the basin was first colonized by stream-adapted fish, and the lake population is derived from those ancestors. The authors here do a fair job of presenting that alternative view in the Introduction and addressing it in the work here. They take multiple perspectives (genetic datasets, demographic models) to address the issues. However, there are a few things that could be added to strengthen the manuscript.

We thank reviewer #3 for constructive feedback on our manuscript.

- mtDNA haplotype and microsat data show a strong signal of association with eastern European/Baltic origins for the Lake Constance populations, with input also from other sources. This aligns with phenotypic data on plate morphs, in which the lake populations are dominated by high-plated fish, as are the freshwater populations in eastern Europe.

R3.1 - Newly generated RAD sequencing data for this study also support this scenario of lake and southeast stream populations grouping with eastern European source populations, but this is only tested on 2 populations very close to each other in the southeast portion of the lake. It is extremely unfortunate that the RAD data from the Roesti et al work is not compatible with this group of authors!

Indeed this is unfortunate, but we were now able to include additional *SbfI*-derived RAD data from one additional stream population (NID) and a lake population (ROM) now overlapping between the two datasets! And by extending the demographic modelling to the source populations from West and East Europe, we

are now able to run the exact same models for all of the lake-stream comparisons. Therefore, the hybrid origin (formerly: hybrid speciation) scenario is now supported for combinations with all five stream populations.

R3.3 - Even though the RAD data are distinct between the datasets, different sets of demographic scenarios should not be tested on each dataset, or else there is not a direct comparison of competing hypotheses. The hybrid speciation and secondary contact models are not well-explained in the text, but HybSpe-M-c seems to have a bottleneck that is not represented in the R tests. These scenarios must be consistent across datasets — if they are and I misunderstand the scenarios, they should be better described.

We fully agree with this criticism and have accordingly revised all demographic modelling analysis with (a) including source populations and overlapping populations (NID/ROM) between datasets, (b) structuring models and inference by complexity into multi- or 2-dimensional site-frequency spectra to alleviate trade-offs between model complexity and limited data, (c) applying exactly the same models to different datasets and data subsets, including scenarios with or without ghost populations. We have also simplified the description of these different scenarios and improved their visualization, and will publish all model and SFS data files as supplementary data on Dryad to ensure full reproducibility.

R3.4 - The evidence for adaptive introgression based on F_{st} is an interesting result. It would be instructive to show the regions of reduced F_{st} between western and stream populations along the genome, and compare to the stream-lake and stream-eastern comparisons. With the focus on lateral plate phenotype, the authors should look at patterns of differentiation around the *Eda* locus in particular.

We have added two supplementary figures showing genome-wide differentiation between Lake Constance populations from South-Eastern tributaries and West and East European populations (Fig. S10-S11). However, we did not focus on at differentiation around the *Eda* locus because lake and stream ecotypes in South-Eastern tributaries are not differentiated in this genomic region and consequently not in their lateral plate morph phenotype (but they differ in plate size, see (Marques *et al.* 2016)).

R3.5 - It is always worth considering multiple values of k in Structure analyses, not just the best-supported one. What do larger values of k show with the microsat data? Perhaps a supplemental figure could show this.

We have added multiple k and both microsat and RAD-SNP clustering analyses to the supplementary figures Fig. S4-S5.

R3.6 - The abbreviations used for demographic models (e.g. lines 174-185) are a bit unclear at first.

We have revised all model abbreviations – the new abbreviations for 5-population models are PD = primary divergence, EV* = ecological vicariance, HO* = hybrid origin, SC* = secondary contact.

R3.7 - Fig 1: Are there multiple haplotypes in the potential source populations that are not found in the Lake Constance basin? The filled circles in part (a), set next to the pie charts in part (b), suggest no, but what is the haplotype diversity in the source populations? And how much haplotype variation is shared among source populations? Also in 1c, the lowest C1 label on the tree should be the same font size as the other C1 labels.

Good points – we have revised Fig. 1 to represent more haplotypes from divergent lineages we explicitly want to exclude as potential sources (e.g. Black Sea or Mediterranean lineages). As pointed out however, there are of course many more haplotypes found across Central European watersheds which are distinct but similar to the ones found in Lake Constance and also part of the central European lineage (Makinen & Merila 2008; Lucek *et al.* 2010; Fang *et al.* 2018). The European mtDNA haplotype diversity has been the focus of the paper by Makinen and Merila (2008). Including the many haplotypes described, we are afraid, would overload Fig. 1. We thus opted for focusing on the European distribution of those haplotypes found in Lake Constance and we refer the reader to the original publication by Makinen and Merila (2008) rather than depicting all of them in Fig. 1.

R3.8 - Table 1: Can references be given directly in the table, rather than letters that are then linked in the caption? Check the reference for NID.

Changed accordingly, now as a supplementary table Tab. S3.

References

- Fang B, Merila J, Ribeiro F, Alexandre CM, Momigliano P (2018) Worldwide phylogeny of three-spined sticklebacks. *Molecular Phylogenetics and Evolution* **127**, 613-625.
- Feder JL, Berlocher SH, Roethele JB, *et al.* (2003) Allopatric genetic origins for sympatric host-plant shifts and race formation in *Rhagoletis*. *Proceedings of the National Academy of Sciences of the United States of America* **100**, 10314-10319.
- Lucek K, Roy D, Bezault E, Sivasundar A, Seehausen O (2010) Hybridization between distant lineages increases adaptive variation during a biological invasion: stickleback in Switzerland. *Molecular Ecology* **19**, 3995-4011.

- Makinen HS, Merila J (2008) Mitochondrial DNA phylogeography of the three-spined stickleback (*Gasterosteus aculeatus*) in Europe-evidence for multiple glacial refugia. *Molecular Phylogenetics and Evolution* **46**, 167-182.
- Marques DA, Lucek K, Meier JI, *et al.* (2016) Genomics of rapid incipient speciation in sympatric threespine stickleback. *PLoS Genetics* **12**, e1005887.
- Roesti M, Kueng B, Moser D, Berner D (2015) The genomics of ecological vicariance in threespine stickleback fish. *Nature Communications* **6**, 8767.
- Wright KM, Lloyd D, Lowry DB, Macnair MR, Willis JH (2013) Indirect evolution of hybrid lethality due to linkage with selected locus in *Mimulus guttatus*. *PLoS Biology* **11**, e1001497.

Reviewers' Comments:

Reviewer #4:

Remarks to the Author:

Prior work on the Lake Constance stickleback revealed a puzzle. Some of the differences between lake and stream forms of stickleback seemed relatively ancient, while recent history documents that the introduction of stickleback into the lake is very recent (only about 100 years or so). The authors describe an analysis of genomic (RADseq) data as well as mtDNA sequence data that resolves this puzzle by suggesting that secondary contact and hybridization between Eastern and Western European lineages led to rapid sympatric divergence partly involving the older geographically divergent variation.

As far as I can understand the analysis, it appears to have been carried out carefully and thoughtfully, and in order to arrive at the conclusions the authors were careful to rule out a large number of alternative hypotheses using, as far as I can see, a demographic analysis based on conversion of highly filtered and massaged SNP data into "multidimensional site frequency spectra". I'm a little unclear what these are, but I suppose the authors have some idea what they are doing. I would say that the elaborateness of the analysis and the very large number of possible hypotheses make one suspect that there might be other elaborate hypotheses in the universe of the possible out there that were not dealt with which may be closer to the truth. I think the authors should make it clear that there are also some simple hypotheses being tested at each stage, and that the major conclusions do not depend on the methods of data analysis.

Most of the results are shown in the figures. However, these figures are very busy and detailed with tiny writing and legends. This makes it often hard to understand the results.

Fig. 1.

For instance, if these are not mentioned again in the manuscript, perhaps the authors might like to abandon the haplotype names (invented in refs 22,25) in Fig. 1a, and use the localities only against the haplotype network. Why a network and not a tree? Surely a mitochondrial network does form a tree? I don't see a FRS4 site or is it part of FRS4 RUD/SEY? Meditertranean is spelt wrong. In fig. 1c, why not use a single tree rather than repeating the backbone multiple times? And here, explain that this is a concatenated tree, which is of course unreliable except as to average tree in the genome.

Line 179: explain this is a concatenated phylogenetic result.

ll. 212-3; hard to understand. Explain that these are Lake Constance lake and stream populations, and which is which.

Fig. 3 and Table 1 are very hard to understand. The little letters in particular: HO_{bi} for instance, and SbfI^{oivln} for instance. I couldn't make much sense of these, though I'm dimly aware they refer to something in Fig. 3. If possible, the authors should try to test simpler hypotheses and maybe use all the data at one go rather than only part of the data as far as I can see.

I had similar problems with fig. 4. How could "admixture proportion in percentage of the target" population be 770% in the first green figure in b? Perhaps it is the actual portion of the target Ne estimate that is estimated to be from this source? Some arrows in this image are so small as to be almost invisible.

Lines 344-5: Fst negatively correlated with nucleotide diversity. Isn't this obviously always true, since

Fst $\sim d_{XY}/\pi$? Rephrase. Incidentally, in Fig. 6a,b the red and blue blobs are very small and hard to read. Instead of SNPs, would not windowed Fst be more powerful. A powerful argument for Fst islands being real and caused by abundant ongoing gene flow is if Fst ~ 0 except in the islands, but with the SNPwise analysis, you'll always see a lot of noisy SNPs that do not obey what longer haplotypes in their genomic region are doing. (Incidentally, I don't agree with Reviewer 2 who says: "The 'genomic islands of speciation' concept has now been discredited numerous times (e.g. Cruikshank and Hahn)"). Fst is very useful, if used properly, as a tool to examine divergence where there are large variations in π in each interacting species across different genomic regions. In Fig. 6c,d I think I get the message but it could be more clearly explained.

II. 418-428. It certainly looks as though these are recombinant populations that have evolved recently, and the evidence from Fig. 6c,d certainly suggest that the lake and stream forms incorporate variable amounts of each source, as expected (especially d) if the lake and stream phenotypes are polygenic.

Does it prove, however, that a single source population could not have generated the lake vs. stream phenotypes in the Constance region? It's the old hybrid speciation problem of proving that the hybridization itself actually led to the ability to diverge. The evidence seems suggestive that it could not, but I wonder!

I. 435-6: "Our analysis of ... microsatellite.... data." Microsatellites? Where are these mentioned? Did I miss something?

I. 453-5: "Discrepancy". The problem with using any kind of modelling of demography with SFS is that you're assuming neutrality where actually massive selection must have been going on in some parts of the genome to cause the divergence, and this may well affect linked SFS.

Due to time constraints, I've not checked the supplementary files.

Bern, June 24, 2019

Detailed responses to referee comments

Reviewer #4

Prior work on the Lake Constance stickleback revealed a puzzle. Some of the differences between lake and stream forms of stickleback seemed relatively ancient, while recent history documents that the introduction of stickleback into the lake is very recent (only about 100 years or so). The authors describe an analysis of genomic (RADseq) data as well as mtDNA sequence data that resolves this puzzle by suggesting that secondary contact and hybridization between Eastern and Western European lineages led to rapid sympatric divergence partly involving the older geographically divergent variation.

We thank reviewer #4 for the constructive feedback on our manuscript.

As far as I can understand the analysis, it appears to have been carried out carefully and thoughtfully, and in order to arrive at the conclusions the authors were careful to rule out a large number of alternative hypotheses using, as far as I can see, a demographic analysis based on conversion of highly filtered and massaged SNP data into "multidimensional site frequency spectra". I'm a little unclear what these are, but I suppose the authors have some idea what they are doing. I would say that the elaborateness of the analysis and the very large number of possible hypotheses make one suspect that there might be other elaborate hypotheses in the universe of the possible out there that were not dealt with which may be closer to the truth. I think the authors should make it clear that there are also some simple hypotheses being tested at each stage, and that the major conclusions do not depend on the methods of data analysis.

We fully agree that there might potentially be model scenarios that would fit our data even better. However, given the complexity of the model space it appears impossible to explore all possible models and there is a danger of overfitting by adding parameters. As suggested by reviewer #4, we have now emphasized that in our approach, we start with simple models and increase the complexity enough to eventually test the major competing ecotype divergence hypotheses (now L218ff). Furthermore, we mention in the discussion that additional model exploration might lead to models that fit the data even better, but this is a general problem in complex data modelling. Future work with more data may necessitate a revision in details, but is unlikely to invalidate the conclusions regarding the relative merits of the alternative speciation scenarios that we contrast here (now L465ff).

It is important to realize that we have shown that very different types of analyses (e.g. phylogenetic reconstruction, demographic modelling, D-statistics, hybrid index, cluster analysis) all lead us to the same conclusion. Our conclusions do, hence, not depend on a single method, giving us confidence in the robustness of our inferences.

Most of the results are shown in the figures. However, these figures are very busy and detailed with tiny writing and legends. This makes it often hard to understand the results.

Fig. 1. For instance, if these are not mentioned again in the manuscript, perhaps the authors might like to abandon the haplotype names (invented in refs 22,25) in Fig. 1a, and use the localities only against the haplotype network. Why a network and not a tree? Surely a mitochondrial network does form a tree? I don't see a FRS4 site or is it part of FRS4 RUD/SEY? Meditertranean is spelt wrong. In fig. 1c, why not use a single tree rather than repeating the backbone multiple times? And here, explain that this is a concatenated tree, which is of course unreliable except as to average tree in the genome.

We removed haplotype names from the figure and instead added a color code to the methods section for full reproducibility / easy connection to earlier literature. We also replaced the haplotype network with a maximum likelihood phylogeny.

Unfortunately, the localities in Fig. 1a and 1c are necessary to understand the switch of population topologies in Fig. 1c. We have therefore left population abbreviations, but we have simplified the tree in Fig. 1c. We decided to retain the single-Constance population trees in Fig. 1c and the single tree in Fig. S1, because the topology switch between NID and S2/ROM/L2 is the most important finding of this phylogenetic analysis.

We have fixed the map typo and clarified that FRS4/RUD/SEY refer to approximately the same site (see also Tab. S3 for details on location and data types associated with each location).

Line 179: explain this is a concatenated phylogenetic result.

We added this information here (now L180) and also at two additional places (now L118, L141, L164).

II. 212-3; hard to understand. Explain that these are Lake Constance lake and stream populations, and which is which.

We revised the sentence along those lines (now L214ff).

Fig. 3 and Table 1 are very hard to understand. The little letters in particular: HO_{bi} for instance, and S_{bfl}^{oivln} for instance. I couldn't make much sense of these, though I'm dimly aware they refer to something in Fig. 3. If possible, the authors should try to test simpler hypotheses and maybe use all the data at one go rather than only part of the data as far as I can see.

We have replaced the subscripts in Table 1 with numbers referring to populations, to avoid confusion with letters in model names. For the 'model name small letters', we have made replacements to avoid confusion and explained what they mean in the legend of Fig. 3 (o = admixture from Rhone, i = admixture from Rhine, x = admixture from both), as well as clarified Fig. 3d where now all modeled scenarios are referenced with full name.

We agree with the recommendation to test simpler models, and this is exactly the reason why we have not used all data at once, but instead started from simple models (with 3 populations) and made them progressively more complex (up to 5 populations). The only possible simpler model with 5 populations would be one in which there is no gene flow between lake and stream ecotypes, which is not biologically realistic and not supported by the data (as estimated migration rates in the more complex models are substantial). Furthermore, testing all data at once is not possible because the two different RAD-seq datasets do not overlap (different restriction enzymes yielding non-overlapping sets of loci). Finally, our strategy of increasing model complexity tries to avoid confounding effects of gene flow between some populations (e.g. within Lake Constance), which further argues against analyzing all data at once, as outlined in the results section (L218ff).

I had similar problems with fig. 4. How could "admixture proportion in percentage of the target" population be 770% in the first green figure in b? Perhaps it is the actual portion of the target Ne estimate that is estimated to be from this source? Some arrows in this image are so small as to be almost invisible.

We thank the reviewer for spotting this error. Indeed the 770 is the number of individuals migrating to the other deme, not a percentage. We converted these numbers back to percentages, in agreement with the figure legend, for both Fig. 4b and 4c. We also increased the size of arrows.

Lines 344-5: F_{ST} negatively correlated with nucleotide diversity. Isn't this obviously always true, since $F_{ST} \sim d_{XY}/\pi$? Rephrase.

We agree that F_{ST} should usually correlate negatively with π , except if F_{ST} is driven by variation in d_{XY} across the genome, e.g. due to differential introgression across the genome, when genomic islands of reduced introgression are also high diversity regions (Cruickshank and Hahn 2012). We have fully revised our discussion of

genome-wide correlations of statistics and Fig. S9 to better emphasize the key result: a divergent selection scenario between lake and stream habitat best explains differentiation landscapes, while the evidence does not suggest an important role of background selection driving differentiation landscapes, even among allopatric populations.

Incidentally, in Fig. 6a,b the red and blue blobs are very small and hard to read. Instead of SNPs, would not windowed F_{ST} be more powerful.

We enlarged colored dots and decreased black dots to make red/blue points better readable in Fig. 6a,b.

A powerful argument for F_{ST} islands being real and caused by abundant ongoing gene flow is if $F_{ST} \sim 0$ except in the islands, but with the SNPwise analysis, you'll always see a lot of noisy SNPs that do not obey what longer haplotypes in their genomic region are doing. (Incidentally, I don't agree with Reviewer 2 who says: "The 'genomic islands of speciation' concept has now been discredited numerous times (e.g. Cruikshank and Hahn)"). F_{ST} is very useful, if used properly, as a tool to examine divergence where there are large variations in π in each interacting species across different genomic regions. In Fig. 6c,d I think I get the message but it could be more clearly explained.

We fully agree that the noisy signal at single SNPs could be smoothed with a windowed F_{ST} approach. However, in Fig. 6, the main purpose was to assign each SNP to either origin (red / blue, West / East), which we attempted by using SNP allele frequencies, an approach that would not work with windows. We have therefore retained the SNP-based analysis. With whole genome data and longer haplotypes available, it would indeed be great to repeat such an analysis in the future, where haplotypes may contain enough information to be assigned to West or East European origin respectively. Unfortunately, this is not possible with RAD-seq data and its associated very large windows that span several rather distant RAD tags interspersed by missing data.

We have rephrased the figure legend of Fig. 6 to convey the message better to the reader.

II. 418-428. It certainly looks as though these are recombinant populations that have evolved recently, and the evidence from Fig. 6c,d certainly suggest that the lake and stream forms incorporate variable amounts of each source, as expected (especially d) if the lake and stream phenotypes are polygenic.

Does it prove, however, that a single source population could not have generated the lake vs. stream phenotypes in the Constance region? It's the old hybrid speciation

problem of proving that the hybridization itself actually led to the ability to diverge. The evidence seems suggestive that it could not, but I wonder!

A very good question – would a single lineage be able to form lake and stream ecotypes, especially on such a short timescale and with persistence in sympatry? Theoretically, a single source population could of course have generated lake vs. stream ecotypes, of which many examples exist across the Northern Hemisphere. Our own study of other lake-stream ecotype pairs in Switzerland that have evolved on similar time scales suggest that ecotypes can indeed evolve from predominantly one source, but phenotypic differentiation is weaker (Lucek et al. 2013 J Evol Biol) and they cannot persist in sympatry (unpublished data), as opposed to Lake Constance ecotypes (Marques et al. 2016 PLOS Genetics).

In our manuscript, we show that at least two source populations admixed in Lake Constance and generated lake and stream ecotypes. We think that the evidence we present – in particular introgression of West European alleles into the opposite genomic background, which led to a new stream ecotype – strongly supports that introgression has at least played a crucial role in facilitating divergence into lake and stream ecotypes and early sympatry.

I. 435-6: "Our analysis of ... microsatellite.... data." Microsatellites? Where are these mentioned? Did I miss something?

Apologies for omitting a reference to the data earlier in the results section. The clustering analysis mentioned in L575ff has been performed with both genome-wide SNPs and microsatellites, the latter allowing us to include a larger number of populations for which overlapping SNP datasets are not available. Both analyses are supporting the same conclusion that Lake Constance stream population GRA showed a predominantly West European origin while the remaining Lake Constance lake and stream populations showed minor admixture from West European origin. Results are shown in Figs. S4 and S5. We have added the description of the data to the results in the description of the clustering analysis section in L214ff.

I. 453-5: "Discrepancy". The problem with using any kind of modelling of demography with SFS is that you're assuming neutrality where actually massive selection must have been going on in some parts of the genome to cause the divergence, and this may well affect linked SFS.

While we agree that such interference cannot be excluded, we have tried to minimize the effects of selection by including only high recombination rate regions into the SNP dataset. Furthermore, selection would not be expected to have a similar effect on other types of analyses, e.g. D-statistics, phylo-geographic reconstruction etc., which all support our findings independently. We now better acknowledge potential

interference of selection on demographic modelling in an additional sentence in the discussion (L471ff).